EMBO
reports

# The lysine-specific methyltransferase KMT2C/MLL3 regulates DNA repair components in cancer

Theodoros Rampias[1] ID, Dimitris Karagiannis[1] ID, Margaritis Avgeris[2] ID, Alexander Polyzos[1], Antonis Kokkalis[1], Zoi Kanaki[1], Evgenia Kousidou[1], Maria Tzetis[3], Emmanouil Kanavakis[3,4], Konstantinos Stravodimos[5], Kalliopi N Manola[6], Gabriel E Pantelias[6], Andreas Scorilas[2] & Apostolos Klinakis[1,*] ID

## Abstract

Genome-wide studies in tumor cells have indicated that chromatin-modifying proteins are commonly mutated in human cancers. The lysine-specific methyltransferase 2C (KMT2C/MLL3) is a putative tumor suppressor in several epithelia and in myeloid cells. Here, we show that downregulation of KMT2C in bladder cancer cells leads to extensive changes in the epigenetic status and the expression of DNA damage response and DNA repair genes. More specifically, cells with low KMT2C activity are deficient in homologous recombination-mediated double-strand break DNA repair. Consequently, these cells suffer from substantially higher endogenous DNA damage and genomic instability. Finally, these cells seem to rely heavily on PARP1/2 for DNA repair, and treatment with the PARP1/2 inhibitor olaparib leads to synthetic lethality, suggesting that cancer cells with low KMT2C expression are attractive targets for therapies with PARP1/2 inhibitors.

**Keywords** DNA repair; epigenetic regulation; KMT2C; PARPi sensitivity
**Subject Categories** Cancer; Chromatin, Epigenetics, Genomics & Functional Genomics; DNA Replication, Repair & Recombination

## Introduction

It is well established that epigenetic dysregulation is an integral component of cancer etiology and progression [1]. Therefore, it is not surprising that numerous epigenetic modifiers, such as DNMT3A, EZH2, and the MLL proteins, are frequently found genetically altered in cancer [2,3]. Lysine (K)-specific methyltransferase

2C (KMT2C, also known as MLL3) belongs to the mixed-lineage leukemia (MLL) family of histone methyltransferases which methylate the histone 3 tail at lysine 4 (H3K4) [4] as part of the complex proteins associated with Set 1 (COMPASS) complex [5]. Although originally identified as oncogenic fusions in leukemia [6], recent genome-wide mutation studies have revealed frequent, presumably loss-of-function, mutations in various members of the MLL family, including MLL2/KMT2B, MLL3/KMT2C, and MLL4/KMT2D in a variety of malignancies, particularly solid tumors [7–11]. Mouse studies have also uncovered a tumor suppressor role for KMT2C in acute myeloid leukemia (AML) [12] and urothelial tumorigenesis [13]. Mechanistic studies of KMT2C in normal cells have focused primarily on its role in enhancer regulation [14,15] by deposition of H3K4me1 marks. Interestingly, recent reports also indicate roles for KMT2C in transcription regulation, which are independent of its H3K4 monomethylation activity on enhancers [16,17]. However, its role in tumorigenesis remains largely undefined.

Bladder cancer is the fifth most common human malignancy and the second most frequently diagnosed genitourinary tumor after prostate cancer [18]. The majority of malignant tumors arising in the urinary bladder are urothelial carcinomas. Superficial carcinoma accounts for approximately 75% of the newly diagnosed cases while the remaining 25% represents muscle-invasive bladder cancer [19]. The latter, often originating from superficial carcinoma, is a life-threatening disease with high metastatic potential. Recent genome-wide studies on superficial and muscle-invasive urothelial carcinoma have indicated that epigenetic regulators, including *KMT2C*, are commonly mutated in both types [11,20]. Here, we show that KMT2C is downregulated in neoplastic tissue in several epithelial cancers including urothelial carcinoma. As expected, *KMT2C* knockdown leads to epigenetic and expression changes. Of interest, genes involved in DNA damage response (DDR) and DNA repair, particularly homologous recombination (HR)-mediated DNA repair, are

1  Biomedical Research Foundation Academy of Athens, Athens, Greece
2  Department of Biochemistry and Molecular Biology, Faculty of Biology, National and Kapodistrian University of Athens, Athens, Greece
3  Department of Medical Genetics, Medical School, "Aghia Sophia" Children's Hospital, National and Kapodistrian University of Athens, Athens, Greece
4  University Research Institute for the Study and Treatment of Childhood Genetic and Malignant Diseases, "Aghia Sophia" Children's Hospital, National and Kapodistrian University of Athens, Athens, Greece
5  First Department of Urology, "Laiko" General Hospital, Medical School, National and Kapodistrian University of Athens, Athens, Greece
6  Laboratory of Health Physics, Radiobiology & Cytogenetics, National Center for Scientific Research (NCSR) "Demokritos", Athens, Greece
   *Corresponding author. Tel: +30 2106597069; E-mail: aklinakis@bioacademy.gr

downregulated. This leads to increased DNA damage and chromosomal instability, highlighted by generation of micronuclei and numerical/regional chromosome losses. In our experiments, cells with reduced *KMT2C* expression are highly dependent on the alternative end-joining (alt-EJ) pathway for repair of double-strand breaks (DSBs), while inhibition of PARP1/2 causes synthetic lethality.

# Results

### KMT2C is downregulated in human epithelial cancers

Mutational data from published studies show that the majority of *KMT2C* mutations cluster within the plant homeodomain (PHD) fingers 1–3 located in the N-terminus of the protein (Catalogue of Somatic Mutations in Cancer—COSMIC). KMT2C PHD fingers 1–3 act as "readers" of the histone methylation status, recognizing monomethylated H3K4 (H3K4me1), while the catalytic Su(var)3-9, Enhancer of zeste, Trithorax (SET) domain, located in the C-terminus, is the "writer" that adds methyl- groups to complete the methylation process [21]. *KMT2C* is commonly mutated in high-grade muscle-invasive urothelial carcinoma [7], in which mutations were recently found equally distributed within the two major subtypes, luminal papillary and basal squamous [11]. Little is known, however, about low-grade/early-stage tumors, including superficial papillomas. To address this issue, we sequenced the N- and C-terminus of the *KMT2C* transcript in tumors and matching normal tissues from a cohort of 72 patients diagnosed with superficial or muscle-invasive urothelial cancer of variable grade [22]. We identified mutations primarily within PHD fingers 1–3 (Fig 1A), which showed no statistical preference with respect to grade and stage (mutations were found in 12/43 high grade vs. 4/29 low grade, and 9/32 invasive vs. 7/40 superficial tumors). Interestingly, a recent study on non-invasive bladder cancer also identified a high frequency (15%) of *KMT2C* likely loss-of-function mutations in non-invasive bladder cancer [20], indicating that KMT2C inactivation might occur early in carcinogenesis. In our mutation detection study, both frameshift and missense mutations were identified, the vast majority of which are predicted to be damaging (Fig 1A and Table EV1). Recently identified missense mutations within the PHD fingers 1–3 have been shown to disrupt the interaction between

KMT2C and BAP1 leading to reduced recruitment of KMT2C to gene enhancers [1]. Our *KMT2C* expression analysis in 104 matched normal/cancer tissue pairs from an expanded bladder cancer patients cohort ($n = 138$; Appendix Tables S1 and S2) revealed that, in comparison with normal tissues, *KMT2C* expression is downregulated in the majority of tumors at both the RNA and protein levels (71/104, $P < 0.001$; Fig 1B and C).

*KMT2C* is mutated in several epithelial cancers [8], implying a general role as a tumor suppressor. To investigate this hypothesis, we performed a meta-analysis on publicly available RNA-seq data from The Cancer Genome Atlas (TCGA) Consortium [23–26]. We found that similarly to bladder cancer (BC), *KMT2C* is downregulated in comparison with normal tissue in colorectal adenocarcinoma (COAD), non-small-cell lung cancer (NSCLC), and head and neck squamous cell carcinoma (HNSCC; Fig 1D). These data indicate that *KMT2C* downregulation is a rather common event in tumorigenesis in several human epithelial tissues. On the other hand, a recent report [27] and our own meta-analysis of non-epithelial cancers with the use of the GEPIA web server [28] indicated that, in comparison with respective healthy tissue, *KMT2C* is expressed at higher levels in glioblastoma multiforme (GBM), brain lower grade glioma (LGG), diffuse large B-cell lymphomas (DLBL), acute myeloid leukemia (AML), and sarcomas (SARC; Appendix Fig S1). This is in agreement with the fact that *KMT2C* truncating mutations account for only 0.6% in these cancer types (2/397, 2/512, 0/41, 3/200, and 2/254 cases, respectively; not shown).

Our meta-analysis of publicly available DNA methylation data [7] obtained from the MethHC database [29] indicates that two Illumina methylation detection probes (cg17322443 and cg19258062) located within a CpG island (chr7:152435133–152437025, assembly GRCh38/hg38, ENCODE) spanning the *KMT2C* proximal promoter are subject to DNA methylation in bladder tumor samples, while remaining methylation-free in normal tissue (Fig EV1A and B), confirming a previously published report [30]. More importantly, the same CpG island within the *KMT2C* proximal promoter is also hypermethylated in tumor samples from COAD, NSCLC, and HNSCC (Fig EV1C). Collectively, these data indicate that both mutational inactivation and transcriptional downregulation via promoter methylation of *KMT2C* might contribute to reduced activity facilitating tumor development in several epithelial cancers.

▶

**Figure 1.  KMT2C downregulation in cancer tissue.**

A  *KMT2C* mutations identified in our study cohort of human bladder cancers. Mutations in red are predicted to be damaging while those in black benign, according to the PolyPHEN-2 algorithm (D and B, respectively, in Table EV1) [95].

B  Comparison of *KMT2C* expression in cancer/healthy matched tissue pairs ($n = 104$) of the study cohort. Expression is presented as log(ratio tumor/healthy) in the *y*-axis. Data obtained from qRT–PCR analysis. *P* value calculated by Wilcoxon signed-rank test.

C  Immunofluorescence (top) and Western blot analysis (bottom) against KMT2C on representative human bladder cancers with variable KMT2C transcript levels: 11th, 4th, 93rd, and 79th percentiles for UCC30, 6, 7, and 29, respectively (Appendix Table S2), from the differential expression analysis of the study cohort. Antibodies against KRT5 or KRT20 were used to stain urothelial cells and DAPI as nuclear counterstain. β-Actin is used as loading control in Western blots. Scale bars indicate 50 μm.

D  Comparison of *KMT2C* expression in human healthy and cancer tissues from bladder cancer (BC, $n = 136$), colorectal adenocarcinoma (COAD, $n = 128$), non-small-cell lung cancer (NSCLC, $n = 341$), and head and neck squamous cell carcinoma (HNSCC, $n = 174$) patients. For NSCLC analysis, separate cohorts from adenocarcinoma and squamous cell carcinoma were combined. Separate analysis of the two NSCLC subtypes (adenocarcinoma and squamous cell carcinoma) yielded the same results. For COAD, the *y*-axis is the log2(ratio tumor/normal) of *KMT2C* expression as assessed with Affymetrix microarray. All expression data were obtained from TCGA through cbioportal.org. *P* values calculated by Mann–Whitney *U*-test. The middle lines inside the boxes indicate the median (50th percentile). The lower and the upper box boundaries represent the 25th percentile and the 75th percentile, respectively. The lower and upper whiskers extend to the lowest and highest values, respectively, within the 1.5× interquartile range (box height) from the box boundaries.

Source data are available online for this figure.

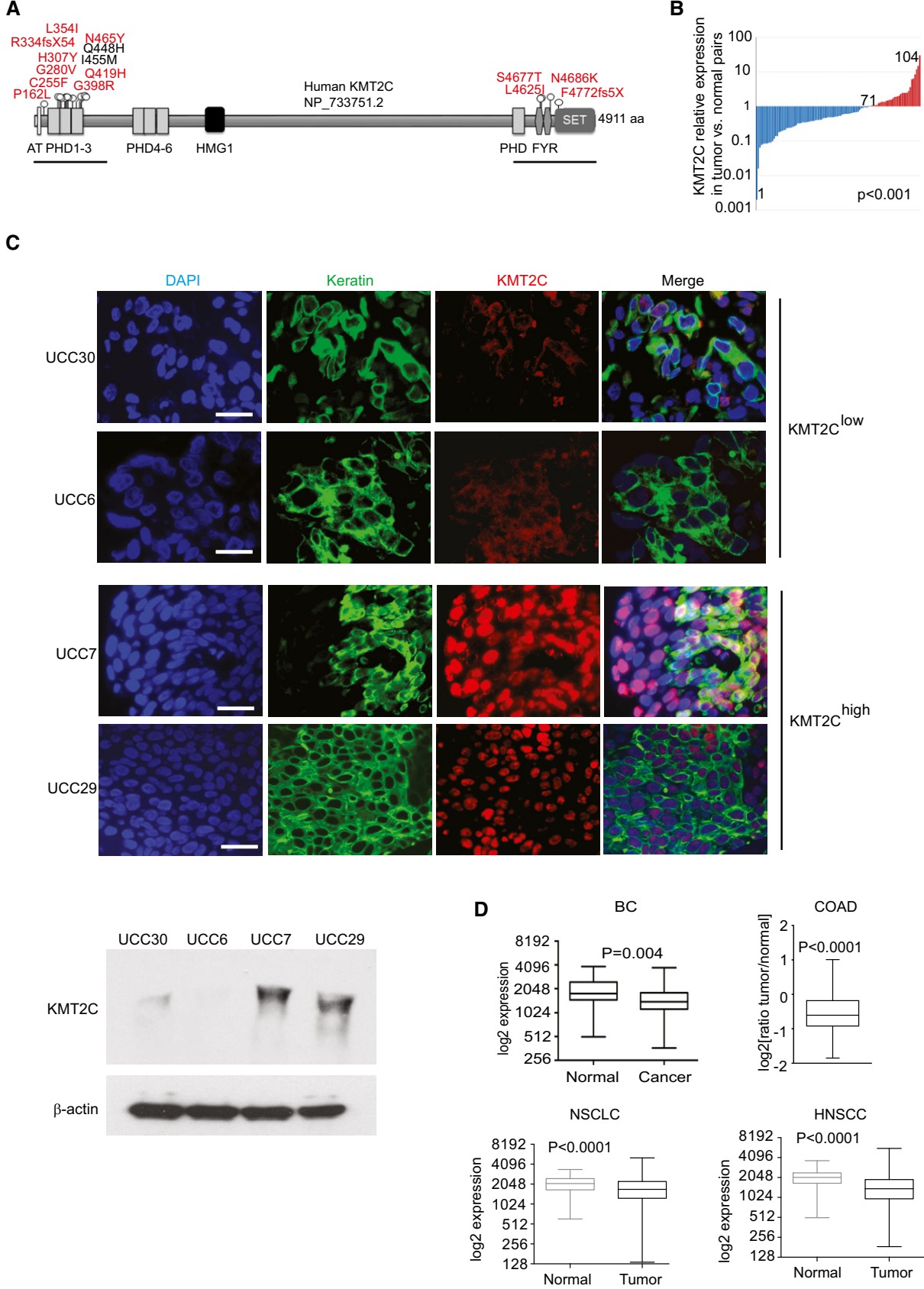

Figure 1.

## KMT2C loss affects enhancer activity and gene expression in a subset of genes

To investigate its role in urothelial carcinoma cells, we used two independent shRNAs (KD1/KD2) to knock down KMT2C levels in human BC cell lines (Fig 2A). While the loss of KMT2C activity did not affect cell proliferation or apoptosis (Appendix Fig S2), RNA-seq experiments in HTB9 cells showed that, directly or indirectly, 3,324 genes were transcriptionally affected upon *KMT2C* silencing (1.4-fold and higher change in expression levels). Of those, 1,846 were downregulated while 1,478 were upregulated. Gene ontology (GO) analysis indicated that many of the affected genes are involved in DDR, DNA repair, DNA replication, cell cycle control, and apoptosis, all of which are considered hallmarks of cancer, and are associated with tumor aggressiveness [31] (Fig 2B). In order to study directly the role of KMT2C and to circumvent the lack of chromatin immunoprecipitation (ChIP)-grade anti-KMT2C antibodies, we exogenously expressed a Flag-tagged KMT2C protein (fKMT2C) in HTB9/KD1 cells (Fig 2C).

To gain further insight into the function of KMT2C in gene transcription regulation, we used fKMT2C-complemented HTB9/KD1 cells to map KMT2C binding genome-wide through ChIP-sequencing (ChIP-seq). In addition, to measure the epigenetic effects of KMT2C loss we performed ChIP-seq experiments for histone 3 lysine 27 acetylation (H3K27ac), histone 3 lysine 4 trimethylation (H3K4me3), and histone 3 lysine 9 acetylation (H3K9ac) histone modifications on HTB9 KMT2C/KD1 and control Scr cells. ChIP-seq experiments performed with anti-Flag antibodies indicated that KMT2C binding sites are equally dispersed among promoter, gene body, and intergenic regions (12,417, 10,882, and 9,885 peaks, respectively; Fig 2D).

In agreement with its role in enhancer regulation, KMT2C colocalizes with the active enhancer mark H3K27ac on intergenic sites likely representing active enhancers [32] (Fig 2E). Our ChIP-Seq analysis identified 2,808 genes proximally located to enhancers that are characterized by KMT2C binding and significant H3K27ac loss upon KMT2C silencing. GO analysis on genes of this group that are also downregulated upon KMT2C silencing (1.5-fold or higher reduction) revealed an enrichment in processes such as focal adhesion and integrin-mediated adhesion as well as ErbB and Wnt signaling pathways (Fig 2F). More specifically, we identified genes that encode proteins which are critical for cell adherence to the epithelial basement membrane: *ITGB1, ITGB6, RHOB*, a putative tumor suppressor also commonly mutated in BC [7,20], *MMP7*; (Fig 2G); the extracellular matrix organization *LOXL2, LOXL4* and *TIMP4*, an epigenetically silenced putative tumor suppressor in bladder carcinoma [33], and epithelial development and differentiation (*SMAD6, SOX2, EREG, WNT11, BMP2*). Interestingly, KMT2D/MLL4 was recently reported to regulate the enhancers of genes involved in cell–cell and cell–matrix adhesion as well as in differentiation of keratinocytes affecting the expression of *ITGB2, ITGB4, LOXL1, LOXL2, SOX7, WNT10A* genes by a similar way [34]. An analysis of transcription factor binding motifs in KMT2C peaks, located at enhancers that are characterized by significant H3K27ac loss upon KMT2C silencing, identified JUNB, TEAD, RUNX1, and MAFA as the most enriched transcription factors (Fig 2H).

## KMT2C localizes at promoters and controls the expression of DNA damage response and repair genes

Interestingly, our ChIP-seq experiments also revealed 12417 fKMT2C binding sites enriched at transcription start site proximal regions (TSS ± 1,500 bp) that contain large domains of H3K4me3 H3K9ac and H3K27ac marks (Fig 3A and B). *KMT2C* silencing was associated with transcriptional suppression of 1,368 genes, which are characterized by promoter-only KMT2C binding. This finding indicates that besides enhancer regulation, KMT2C is also involved in promoter activation in cancer cells. Transcription factor binding motif analysis of fKMT2C-bound regions yielded a totally different set of transcription factors from those identified in enhancers. The most prominent of these is ELK1 (Fig 3C), a prominent RAS/MAPK target controlling components of the basal transcriptional machinery, the spliceosome and the ribosome [35].

Our ChIP-seq and RNA-seq data indicated that upon *KMT2C* silencing, the subgroup of genes showing reduced expression levels also show reduced H3K4me3 levels at the respective TSSs (Fig 3D). GO analysis on this group of the 1,368 downregulated genes revealed several processes such as DDR and DSB repair by HR, which interestingly presented the highest score (see also Fig 2B). More specifically, *KMT2C* silencing was associated with decreased expression of key components of DDR (*ATM, ATR*) and the HR DNA repair pathway (*BRCA1, BRCA2, RAD50, RAD51*; Fig 3E). Interestingly, restoration of KMT2C activity by means of exogenous expression of fKMT2C also restored the expression levels of these genes (Fig 3F).

Our own ChIP-seq data as well as ENCODE data indicate that KMT2C and the COMPASS complex component RBBP5 colocalize together with ELK1 upon the TSS of *ATM, ATR, BRCA1,* and *BRCA2* genes (Fig 4A). Moreover, KMT2C levels modulate positively the H3K4me3 enrichment on TSS of these genes, indicating an important role for this histone methyltransferase on their transcriptional activation. More specifically, upon *KMT2C* silencing, H3K4me3 levels were significantly reduced, whereas restoration of KMT2C activity also restored H3K4me3 levels. Promoter region immunoprecipitation either as direct binding or through long-range enhancer interactions has previously been reported for both KMT2C and KMT2D [36,37]. KMT2C binding upon the promoter region of the *ATM, ATR, BRCA1,* and BRCA2 genes is independently corroborated in a recently published analysis [38] (Appendix Fig S3). Interestingly in the same study, a 32% of KMT2C is located within promoter regions, indicating roles for KMT2C besides enhancer H3K4 monomethylation.

This observation prompted us to knock down *KMT2C* expression in a wide panel of BC, COAD, HNSCC, and NSCLC cell lines which according to publicly available data showed variable KMT2C expression levels (Fig 4B). Quantitative RT–PCR experiments revealed an invariable downregulation of DDR and HR components (Fig 4C). Finally, expression analysis of our cohort of bladder cancer tumors (Fig 4D), as well as meta-analysis of publicly available TCGA expression data from BC, COAD, NSCLC, and HNSCC, indicated that *KMT2C* levels strongly correlate with the expression of the same genes (Fig 4E). Interestingly, a positive correlation between *ATM, ATR, BRCA1, BRCA2, and KMT2C* expression is also derived from TCGA data GBM, LGG, AML, DLBL, SARC, and breast invasive carcinoma (BRIC) RNA-seq data

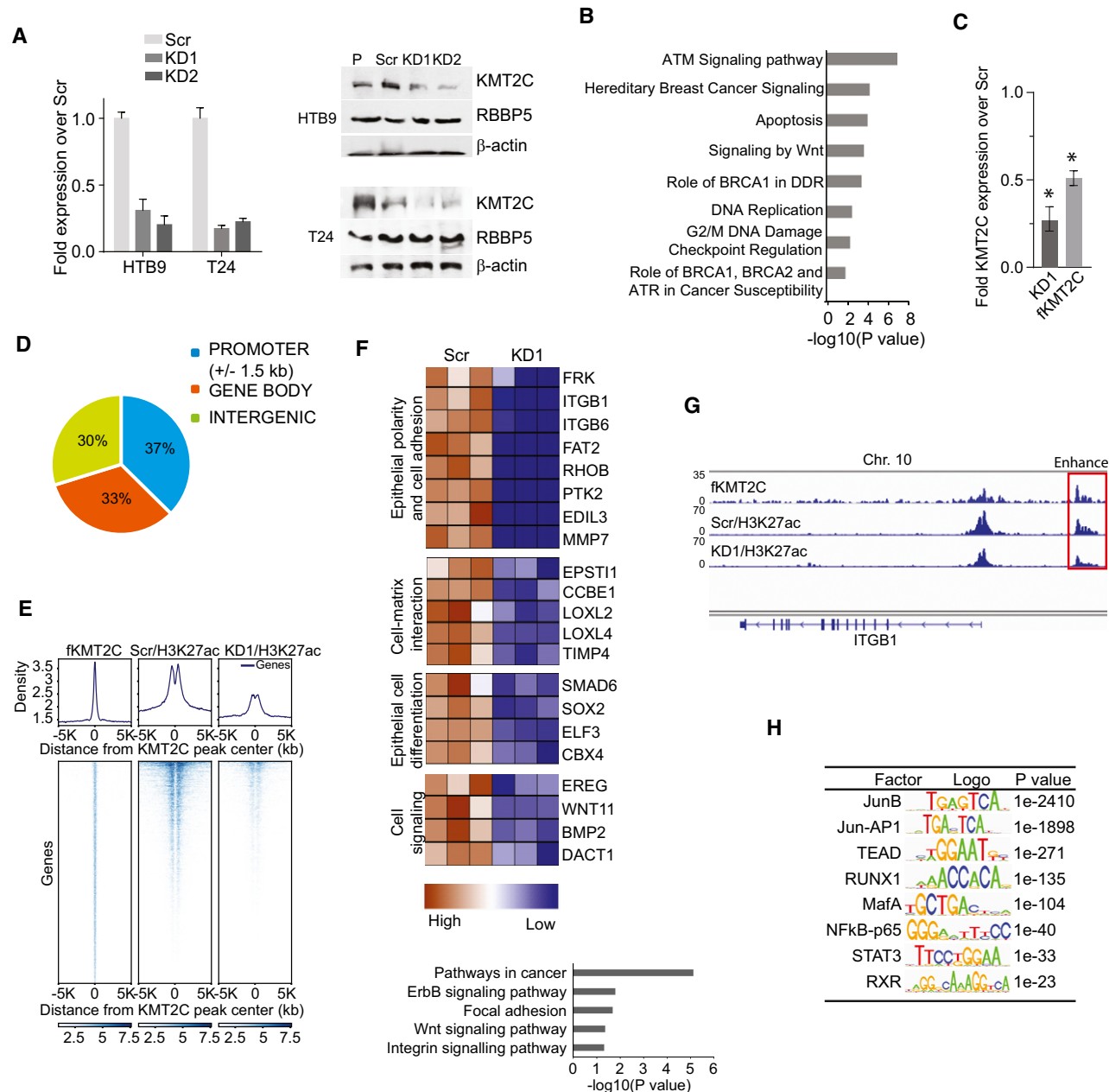

**Figure 2.  KMT2C loss leads to extensive epigenetic changes in human bladder cancer cells.**

A   KMT2C transcript (left) and protein (right) levels in human bladder cancer cell lines stably transduced with lentiviral vectors expressing shRNAs against KMT2C (KD1 and KD2) in comparison with Scr control cells expressing scrambled shRNAs (Scr). RBBP5, another COMPASS complex protein used as internal control and b-actin as loading control. Transcript levels were assessed by qRT–PCR in triplicates, and values shown represent mean ± SEM.

B   Bar graph showing selected biological processes and signaling pathways obtained from Gene Ontology (GO) enrichment analysis for the 3,324 differentially expressed genes between Scr control and KMT2C/KD1 HTB9 cells. Expression values were obtained from RNA-seq data.

C   Quantitative RT–PCR for *KMT2C* in HTB9/KD1 cells, and HTB9/KD1 cells stably transfected with a plasmid expressing a Flag-tagged full-length KMT2C protein (fKMT2C). Expression levels are shown in the *y*-axis as respective ratios over *KMT2C* expression in Scr control cells (Scr expression corresponds to 1). Experiments were performed in triplicates and analyzed with Mann–Whitney *U*-test. Values shown represent mean ± SEM. * designates *P*-value < 0.05.

D   Genome distribution of KMT2C peaks in HTB9/KD1 cells complemented with fKMT2C. Data obtained from ChIP-seq experiments.

E   Density plot indicating KMT2C binding and H3K27ac levels on active enhancers in Scr control and KD1 HTB9 cells.

F   Bar graph showing selected biological processes and signaling pathways obtained from Gene Ontology (GO) enrichment analysis for 253 genes in proximity to active enhancers affected by *KMT2C* knockdown and heatmap of their expression (> 1.5-fold H3K27ac and mRNA downregulation). Data obtained from ChIP-seq and RNA-seq experiments.

G   Bedgraph indicating KMT2C binding and H3K27ac at a putative enhancer of the *ITGB1* locus before and after KMT2C knockdown in HTB9 cells.

H   Transcription factor binding motif analysis on active enhancers affected by *KMT2C* knockdown. Data obtained from ChIP-seq experiments.

Source data are available online for this figure.

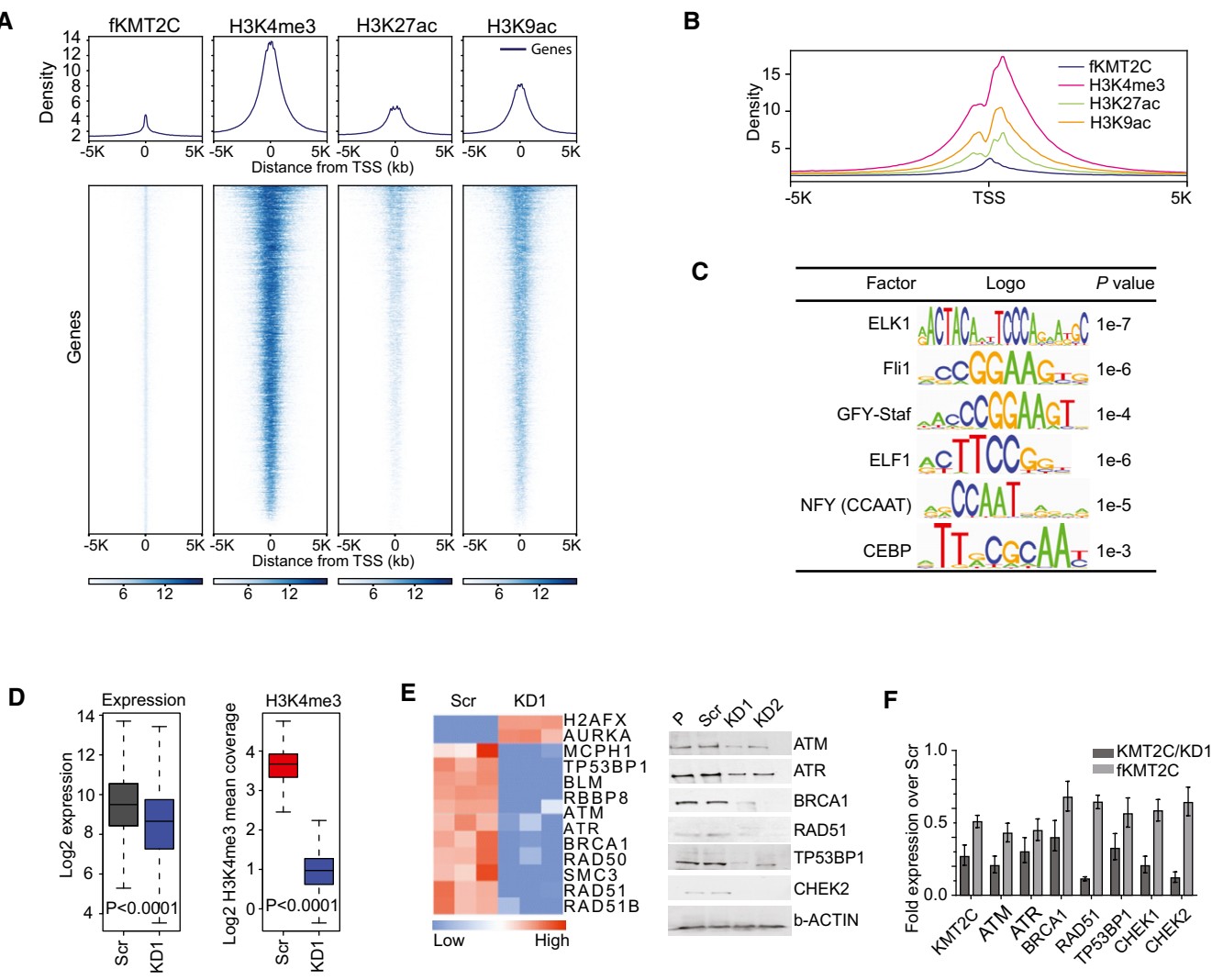

**Figure 3.  KMT2C controls the expression of DDR and DNA repair genes in BC cells.**

A  Density plot indicating KMT2C binding and H3K4me3, H3K27ac, and H3K9ac levels on transcription start sites (TSS) in HTB9 cells.

B  Histogram indicating distribution of histone modifications around transcription start sites (TSS ± 5,000 bp). Data obtained from ChIP-seq with antibodies against the indicated histone modifications.

C  Transcription factor binding motif analysis on TSS of genes transcriptionally affected by *KMT2C* knockdown. Data obtained from ChIP-seq experiments.

D  Boxplot indicating expression (left) and H3K4me3 levels (right) of genes with KMT2C presence on their promoters in Scr control and KMT2C/KD1 cells. Median comparison of Expr/K4 m3 values was performed with two-tailed paired Wilcoxon rank sum test with continuity correction. The middle lines inside the boxes indicate the median (50th percentile). The lower and the upper box boundaries represent the 25th percentile and the 75th percentile, respectively. The lower and upper whiskers extend to the lowest and highest values, respectively, within the 1.5× interquartile range (box height) from the box boundaries.

E  Heatmap comparison of the expression levels of genes implicated in DDR between control (Scr) and *KMT2C*-knockdown (KD1) HTB9 cells (left). Expression data were obtained from RNA-seq experiments. Western blot analysis of selected proteins in control (Scr) and *KMT2C*-knockdown (KD1 and KD2) HTB9 cells (right).

F  Expression level restoration of selected genes in KMT2C/KD1 HTB9 cells complemented with exogenously expressed Flag-tagged KMT2C (fKMT2C). Data obtained by qRT–PCR. Experiments were performed in triplicates, and values shown represent mean ± SEM.

Source data are available online for this figure.

(Appendix Fig S4). Altogether, these data indicate that KMT2C controls the epigenetic status of genes involved in DDR and DNA repair and directly or indirectly their expression levels, even in tissues in which a tumor suppressor role of KMT2C has yet to be established. Deficiencies in DNA repair due to germline or somatic mutations is a common event in cancer [39], while reduced expression of DNA repair components due to epigenetic control, primarily DNA methylation, is also observed [40,41]. In this case,

however, loss of KMT2C seems to affect *en bloc* the expression of multiple key components of the DDR and DNA repair pathways.

**Bladder cancer cells lacking KMT2C are HR-deficient and present high levels of genomic instability**

The observation that KMT2C loss affects genes involved in DDR and DNA repair prompted an in-depth cytogenetic analysis which

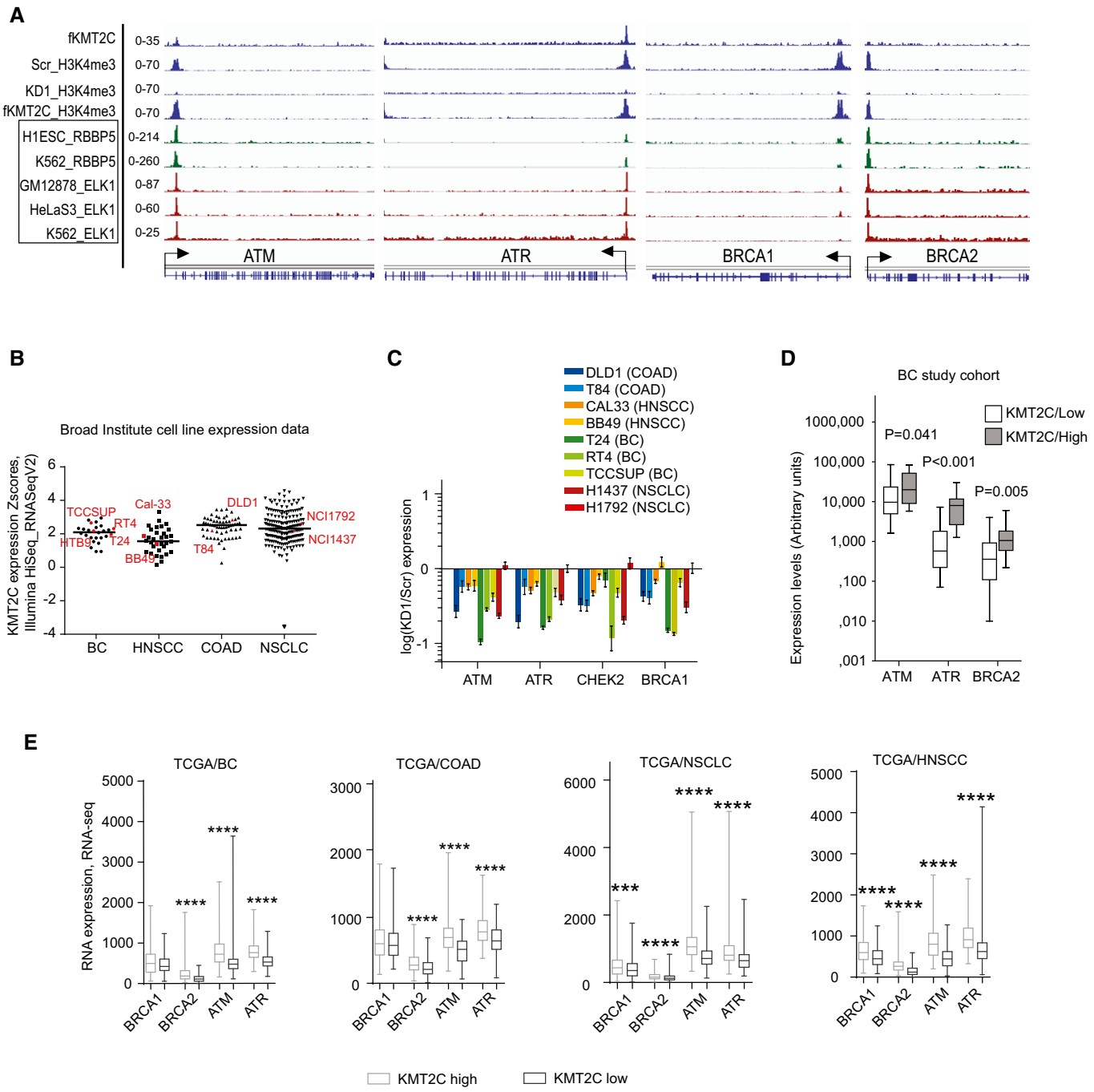

**Figure 4. KMT2C controls the expression of DDR and DNA repair genes in various cancers.**

A  Bedgraphs indicating KMT2C and H3K4me3 binding at the TSS of indicated loci in HTB9 cells; also, from published studies available at the ENCODE, the binding of the COMPASS complex member RBBP5 and the transcription factor ELK1 is indicated in the same loci.

B  *KMT2C* expression (*y*-axis) in various human cell lines (*x*-axis). Cell lines under study are indicated as red geometrical schemes. Data were obtained directly from the Broad Institute CCLE server.

C  Expression levels of indicated genes in indicated cell lines upon *KMT2C* knockdown. Expression is shown as log(KD1/Scr) in the *y*-axis. Remaining KMT2C transcript levels for all knockdown experiments can be found in Table EV2. Note that H1792 which shows poor KD1 (~25%) also shows no change in ATM, ATR, BRCA1, and BRCA2 expression (light red bar appearing last in each set). Experiments were performed in triplicates. In plots, bars represent mean ± SEM.

D  Correlation in expression levels between *KMT2C* and indicated genes in our study cohort of superficial and muscle-invasive BC. Data obtained from qRT–PCR. Experiments were performed in duplicates. *P* values were calculated by Mann–Whitney *U*-test.

E  Correlation in expression levels between *KMT2C* and indicated genes in BC, COAD, NSCLC, and HNSCC tumors. RNA-seq data were obtained from the TCGA through cbioportal.org. Mann–Whitney *U*-test was used. *** designates *P*-value < 0.001 and ****P*-value < 0.0001.

revealed that both HTB9 and T24 cells lacking KMT2C show increased DNA damage as indicated by higher frequency of nuclear foci staining for the DNA damage marker γH2AX. DNA damage levels are comparable to those measured in *BRCA1*-knockdown cells (Figs 5A and EV2A). Cisplatin is known to cause DSBs which, in cells beyond the G1 phase, are repaired by the HR machinery. To assess the contribution of HR in DSB repair, we treated HTB9 and T24 cells with cisplatin and immunostained against γH2AX and the HR repair protein RAD51. While both Scr control and KD cells showed the same frequency of γH2AX foci, RAD51-positive nuclei were significantly fewer in the latter (Figs 5B and EV2B). Moreover, sister chromatid exchange (SCE) assays upon cisplatin treatment clearly demonstrated that, while Scr control cells are HR competent, their KD counterparts show low levels of HR DSB repair (Figs 5C and EV2C). It is known that HR factors stabilize stalled forks by protecting them from nucleolytic degradation, help restarting DNA synthesis from stalled forks, and repair DSBs generated by collapsed forks [42–46]. To investigate the ability of KMT2C/KD cells to resolve stalled forks in S phase, we used DNA fiber assays. In the presence of the DNA replication inhibitor hydroxyurea (HU), KMT2C/KD cells show a behavior similar to that of BRCA1-deficient cells, i.e., inability to resolve stalled forks (Figs 5D and EV2D). These functional data indicate that loss of KMT2C leads to HR deficiency due to downregulation of multiple HR components, as well as compromisation of DNA replication under genotoxic stress.

Interestingly, stalled or collapsed replication forks are a major source of DSBs and endogenous genomic instability in dividing cells [47]. In cancer, oncogene-induced replication stress contributes critically to DNA damage while cancer cells with HR deficiency are characterized by extensive genomic instability [48,49]. Chromosomal instability, as a type of genomic instability, has been also linked to HR deficiency and mitotic defects [50,51]. In KMT2C/KD cells, the increased frequency of micronuclei, chromosome bridges, lagging chromosomes, and chromosome congression (Fig 6A and B) implies gross defects in mitotic fidelity and genome integrity safeguarding. To assess the chromosomal status of KMT2C/KD cells, we used gross karyotypic analysis which revealed that both HTB9 and T24 KMT2C/KD cells had lower chromosomal count in total and per individual chromosome (Fig 6C and D). Moreover, chromosomal microarray analysis (CMA) on HTB9 cells indicated that chromosomal losses were more frequent and more extensive than respective gains (Fig 6E and F).

## KMT2C loss leads to PARP1/2 dependence for DNA repair

Our expression and cytogenetic data clearly indicate that the HR repair machinery is compromised in KMT2C/KD cells. HR deficiency is known to skew the balance toward canonical nonhomologous end joining (c-NHEJ) with the participation of the TP53BP1 protein [52]. However, the number of TP53BP1 foci in cisplatin-treated KMT2C/KD cells is comparable with Scr control cells (Figs 7A and EV3A), possibly implying that the activity of c-NHEJ is not elevated. To this direction, we compared the activity of NHEJ pathway between KMT2C/KD1 and Scr control cells by counting chromosomal fusion events in a dicentric assay. Repair of DSBs induced by ionizing radiation (IR) in this assay generated chromosomal fusions with equal frequency between KMT2C/KD1 and Scr control cells (Figs 7B and EV3B), implying that both employ non-HR mechanisms for DNA repair equally. It is widely accepted that ligase IV participates in the final stages of the c-NHEJ [53], while poly(adenosine diphosphate [ADP]–ribose) polymerase 1 (PARP1) is an integral component of the alt-EJ pathway [54,55]. To assess the individual contribution of c-NHEJ and alt-EJ in DNA repair, we induced DSBs via IR in both Scr control and KMT2C/KD1 cells and measured the frequency of chromosomal fusions in the presence of the ligase IV inhibitor SCR7 [56] or the PARP1/2 inhibitor olaparib [57]. Inhibition of PARP1/2 in KMT2C/KD cells led to a significant reduction ($P < 0.01$) in the number of chromosome fusions, while ligase IV inhibition had a lesser effect (Figs 7C and EV3C). This implies that KMT2C/KD cells rely heavily on alt-EJ for DSB repair. PARP1/2 inhibition in BRCA-deficient cells is known to lead to accumulation of chromosome fragments and radial structures, a phenotype associated with c-NHEJ [52,58,59]. As Fig 7D indicates, blocking the alt-EJ pathway with olaparib leads to significantly ($P = 0.021$) more radial chromosomes in comparison with Scr controls, while simultaneous treatment with olaparib and SCR7 ameliorated this phenotype. Comparable results were obtained when the c-NHEJ and alt-EJ pathways were genetically inhibited through shRNA knockdown of ligase IV and ligase III, respectively (Fig EV4A and B). This further supports the hypothesis that, in KMT2C/KD cells, the alt-EJ pathway plays a more important role in DSB repair.

## Tumors with KMT2C loss are sensitive to PARP1/2 inhibition

Previous reports have shown that HR-deficient cells are sensitive to PARP1/2 inhibitors [58,60]. In fact, recently published results from

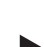

**Figure 5.   Cells lacking KMT2C are HR-deficient.**

A   Immunofluorescence of γH2AX foci (left) and quantitation (right) in control (Scr) and *KMT2C*-knockdown (KMT2C/KD1 and KD2) HTB9 cells. *BRCA1* knockdown (BRCA1/KD) is used as control. The *y*-axis indicates added percentage of cells with 1–5 and > 5 foci for each cell type. Scale bars indicate 5 μm. All comparisons have been performed against Scr control cells. Values in the bargraph represent mean ± SEM from three experiments. Student's *t*-test was used. * designates *P*-value < 0.05, and ** designates *P*-value < 0.01.

B   Frequency of RAD51 foci in cisplatin-treated HTB9 control (Scr) and *KMT2C*-knockdown (KD1) cells. The *y*-axis indicates added percentage of cells with 1–3 and 3 foci. Scale bars indicate 10 μm. Values in the bargraph represent mean ± SEM from three experiments. Student's *t*-test was used. * designates *P*-value < 0.05, and ** designates *P*-value < 0.01.

C   Sister chromatid exchange (SCE) assay with cisplatin-treated HTB9 control (Scr) and *KMT2C*-knockdown (KD1) cells. Red arrowheads indicate sister chromatid exchange events. Results were obtained from 15 metaphases per group. Mann–Whitney *U*-test was used.

D   DNA fiber assay on control (Scr) and *KMT2C*-knockdown (KD1) HTB9 cells. *BRCA1*-knockdown cells are used as controls. Experiments performed with or without hydroxyurea (HU) treatment under the conditions indicated in the schematic. Examples of DNA fibers from HTB9/KD1 cells are shown. The length of minimum 100 fibers from each condition was measured. Values in the plot are means ± SEM. Mann–Whitney *U*-test was used. ** designates *P*-value < 0.01.

    

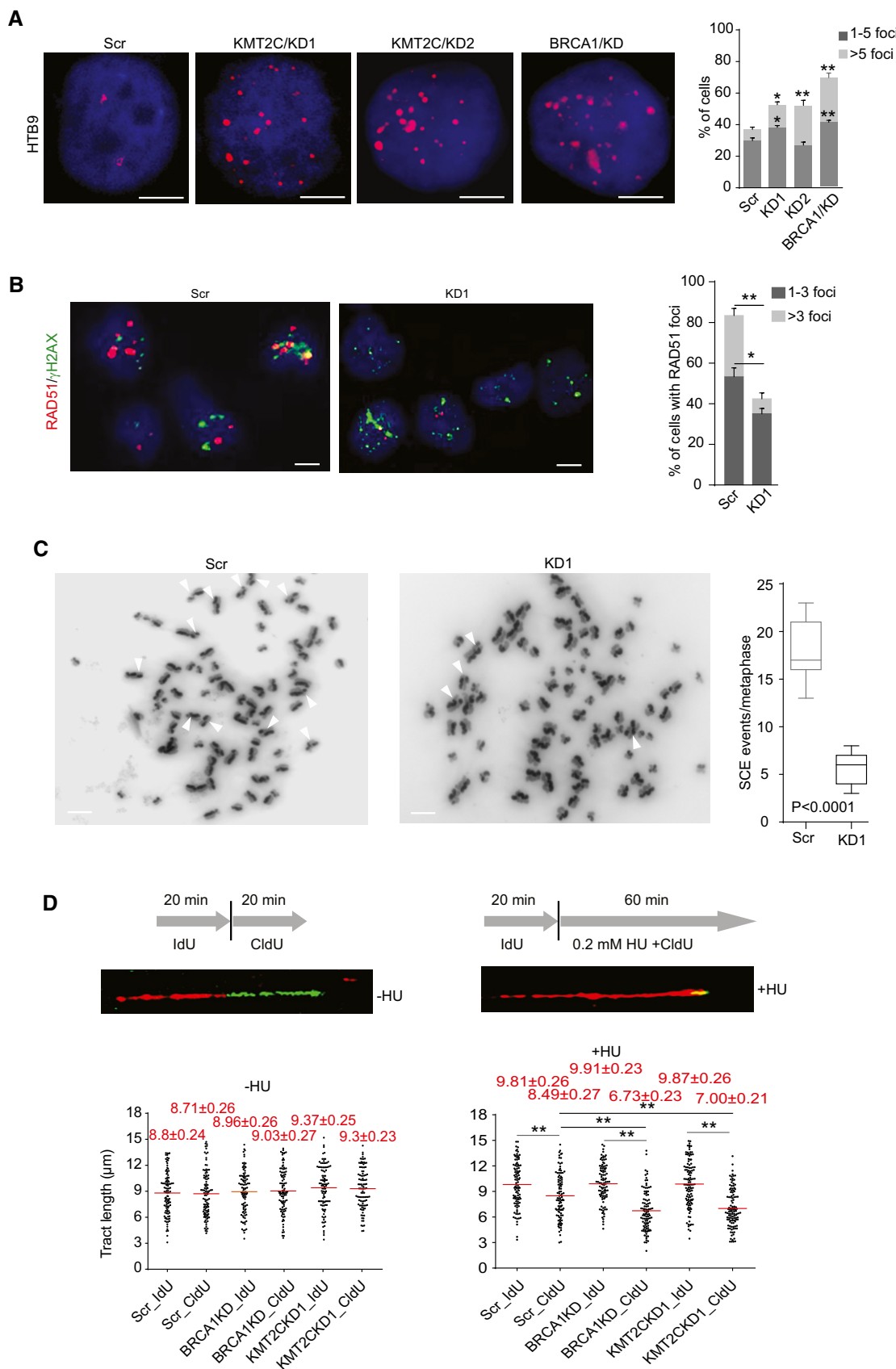

**Figure 5.**

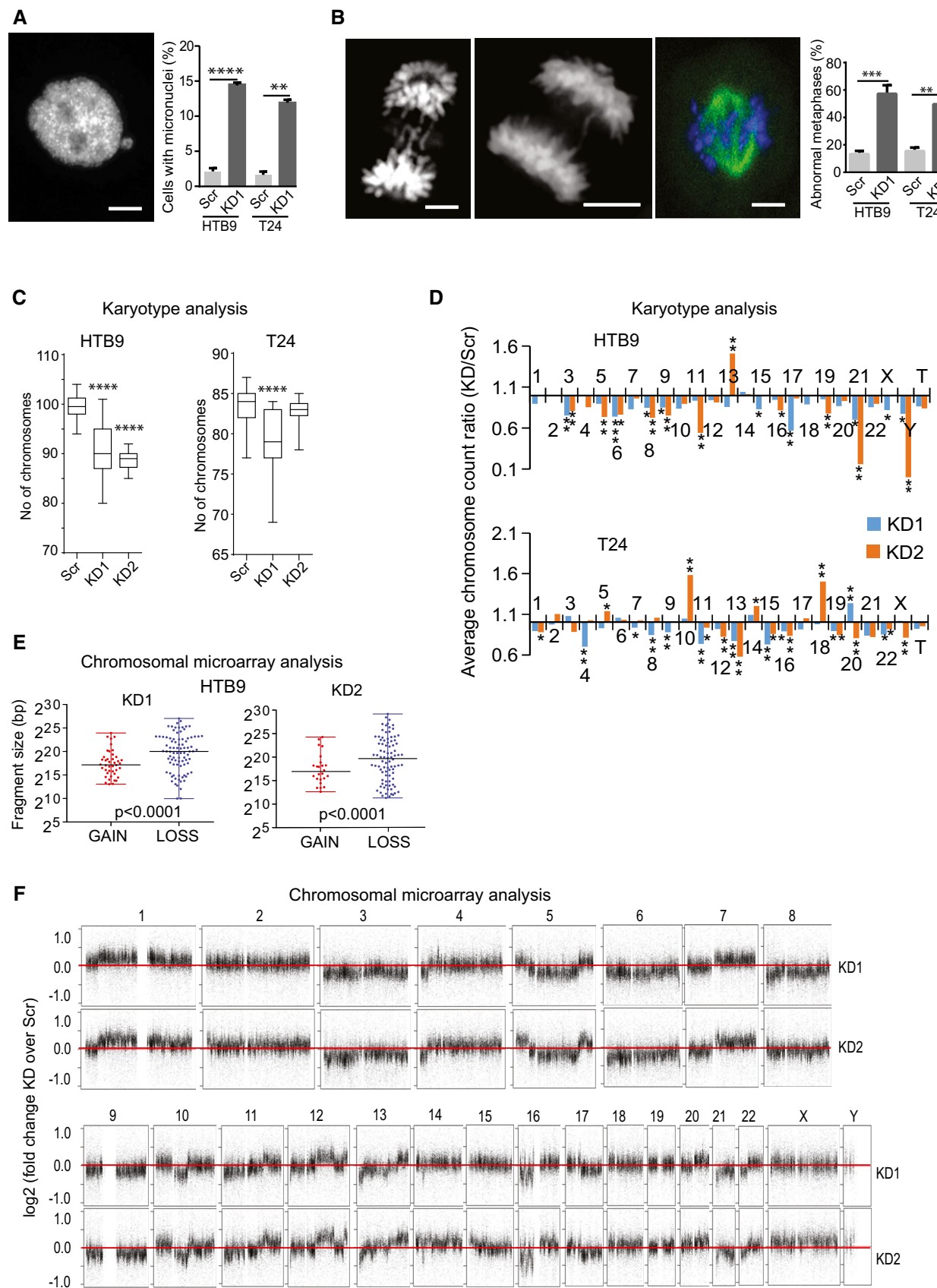

**Figure 6.**

**Figure 6.  KMT2C loss leads to genomic instability.**

A    Representative image (from HTB9/KD1 cells) and frequency of both HTB9 and T24 control (Scr) and KMT2C/KD1 cells with micronuclei. Scale bar indicates 5 μm. Values represent mean ± SEM from three experiments. Student's *t*-test was used. ** designates *P*-value < 0.01 and ****P*-value < 0.0001.

B    Representative images (from HTB9 cells) and frequency of abnormal metaphases presenting lagging chromosomes, chromosome bridges, and chromosome congression in HTB9 and T24 control (Scr) and KMT2C/KD1 cells. Scale bar indicates 5 μm. Values represent mean ± SEM from three experiments. Student's *t*-test was used. **designates *P*-value < 0.01 and *** *P*-value < 0.001.

C    Karyotypic analysis and chromosomal count in control (Scr) and KMT2C/KD1 and KD2 cells. All comparisons performed against Scr control cells. Metaphases studied: HTB9 Scr, *n* = 20; HTB9 KD1, *n* = 11; HTB9 KD2, *n* = 12; T24 Scr, *n* = 20; T24 KD1, *n* = 15; T24 KD2, *n* = 12. Mann–Whitney *U*-test was used. **** designates *P*-value < 0.0001.

D    Average chromosome count ration KD/Scr obtained from Giemsa-stained metaphase spreads of HTB9 and T24 KD1 and KD2 cells. Number of metaphases studied: HTB9 Scr, *n* = 20; HTB9 KD1, *n* = 11; HTB9 KD2, *n* = 12; T24 Scr, *n* = 20; T24 KD1, *n* = 15; T24 KD2, *n* = 12. Mann–Whitney *U*-test was used. * designates *P*-value < 0.05, **P*-value < 0.01, and ***P*-value < 0.001.

E    Fragment size of gains and losses obtained from CMA on HTB9/KD1 and KD2 cells. Mann–Whitney *U*-test was used. One sample from each cell type was used in CMA.

F    Copy number gains and losses of HTB9 KD1 and KD2 cells in comparison with HTB9/Scr controls. Data obtained from CMA. Values as presented as log(KD1/Scr) in the *y*-axis. The horizontal red line indicates log value 0, which corresponds to no change. Note that for the majority of chromosomes there are losses in the KD1 cells.

a Phase II clinical trial indicated that patients with castration-resistant prostate cancer that carry mutations in DNA repair genes, such as *BRCA1/2*, *ATM*, Fanconi anemia components, and *CHEK2*, show positive response to olaparib, indicating a dependence on PARP1/2 for survival upon DNA damage [61]. In agreement with this, KMT2C/KD cells show increased sensitivity to olaparib (Fig 8A). This observation was also confirmed in long-term treatments in clonogenic assays with three different concentrations of olaparib (Fig EV5). Moreover, generation of DSBs through IR is detrimental for olaparib-treated KMT2C/KD cells, underscoring the dependence of these on PARP1/2 for DNA repair (Fig 8B). These findings are independently corroborated from publicly available data from the cancerrxgene.org database [62], which show that cell lines from BC, NSCLC, HNSCC, and COAD with reduced expression of KMT2C are more sensitive to PARPi (Fig 8C).

To corroborate our finding *in vivo*, we used one cell line from each tumor type under study in xenograft experiments (HTB9/BC, H1437/NSCLC, T84/COAD, and Cal-33/HNSCC). Although T84 and Cal-33 KMT2C/KD cells grew somewhat slower *in vivo* in comparison with the respective Scr controls, KD1 cells are more sensitive to olaparib which totally suppressed tumor grown in mice (Fig 8D). This was associated with reduced proliferation, high DNA damage, and severe apoptosis in KD1 cells in all cell lines tested (Fig 8E). Altogether, our *in vitro* and *in vivo* experiments indicate that KMT2C/KD cells rely heavily upon alt-EJ for DSB repair. Although we cannot exclude alt-EJ-independent PARP1/2 functions in DNA repair, we hypothesize that upon inhibition of alt-EJ, KMT2C/KD cells rely exclusively upon c-NHEJ for DSB repair. This, however, is either insufficient or too error prone to deal with elevated DNA damage, eventually leading to cell death.

## Discussion

Histone-modifying enzymes have emerged as critical players in tumor biology in recent years. H3K4 methyltransferases have been implicated in tumorigenesis as both oncogenes and tumor suppressors in a variety of neoplasias. Bladder cancer presents some of the highest reported mutation rates in *KMT2C* and *KMT2D*, and to a lesser extent in *KMT2B* [7,11,63]. A high percentage of reported mutations lead to truncated protein products with presumably impaired functionality. Recent reports, however, indicate that loss of the catalytic activity of KMT2C and KMT2D has a less severe effect on transcription regulation than the respective complete gene knockout [16,17], implying that these proteins may have additional roles in transcriptional regulation beyond H3K4 monomethylation.

In support of this, loss of the catalytic activity of the *Drosophila* homolog *Trithorax* (*Trr*) has negligible effect on fly development, while its complete loss leads to embryonic lethality [64,65]. Therefore, somatic mutations, even those truncating the protein from its catalytic activity, might not be the only MLL-related genetic events associated with cancer. These observations prompted us to focus our studies on the expression levels of KMT2C and its role in already transformed cells. We report for the first time that the epigenetic regulator KMT2C is significantly downregulated in many different types of cancer. We thus speculate that loss-of-function mutations in combination with progressively reduced gene expression due to promoter methylation limit KMT2C activity in cancer cells. Thus, in tumor evolution, promoter methylation of *KMT2C* may provide a selective advantage to emerging *KMT2C*-mutated cells by reducing wild-type protein levels. In support of this model, *KMT2C* mutations were recently identified as late events in

**Figure 7.  KMT2C loss leads to PARP1/2 dependence for DNA repair.**

A    Frequency of TP53BP1 foci in cisplatin-treated HTB9 control (Scr) and *KMT2C*-knockdown (KD1) cells. Size bars in microscopy panels correspond to 5 μm. In the plot, bars represent mean ± SEM from *n* = 3 experiments.

B    Frequency of chromosome fusions obtained from IR-treated (schematic) HTB9 control (Scr) and *KMT2C*-knockdown (KD1) cells. Representative karyotypes are shown. Size bars in karyotype panels correspond to 10 μm. White arrows indicate chromosome fusion events. In the plot, bars represent mean ± SEM from *n* = 3 experiments.

C    Frequency of chromosome fusions in IR-treated HTB9 control (Scr) and *KMT2C*-knockdown (KD1) cells upon treatment with SCR7 (30 μM) and olaparib (15 μM). Bars represent mean ± SEM from *n* = 3 experiments. *** designates *P*-value < 0.001.

D    Frequency of radial chromosomes in IR-treated HTB9 control (Scr) and *KMT2C*-knockdown (KD1) cells upon treatment with SCR7 (30 μM), olaparib (15 μM), or both. Representative karyotypes are shown. Size bars in karyotype panels correspond to 10 μm. White arrows indicate radial structures. In the plot, bars represent mean ± SEM from *n* = 3 experiments. * designates *P*-value < 0.05 and ***P*-value < 0.001.

Throughout the figure, Mann–Whitney *U*-test was used.

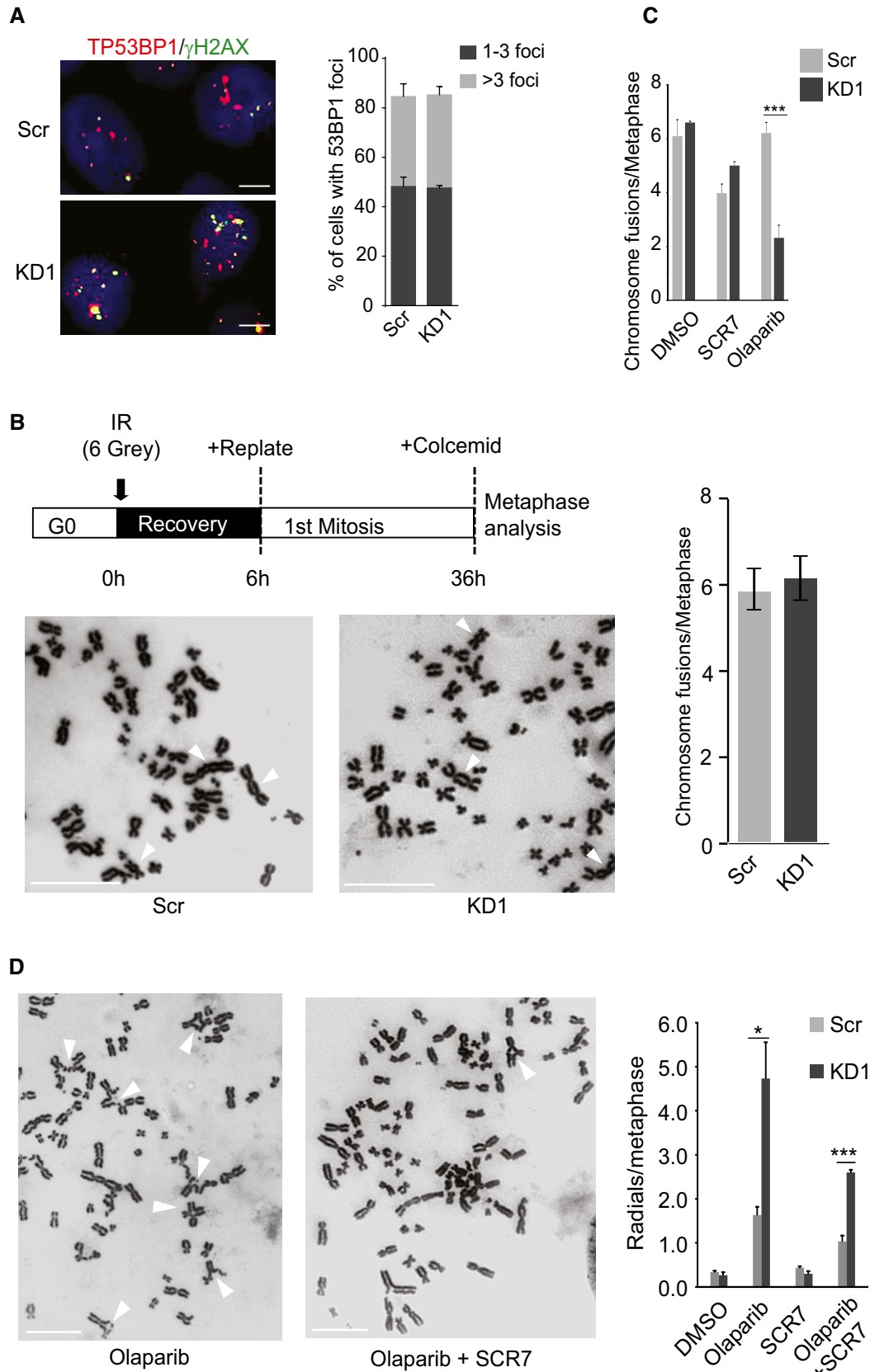

Figure 7.

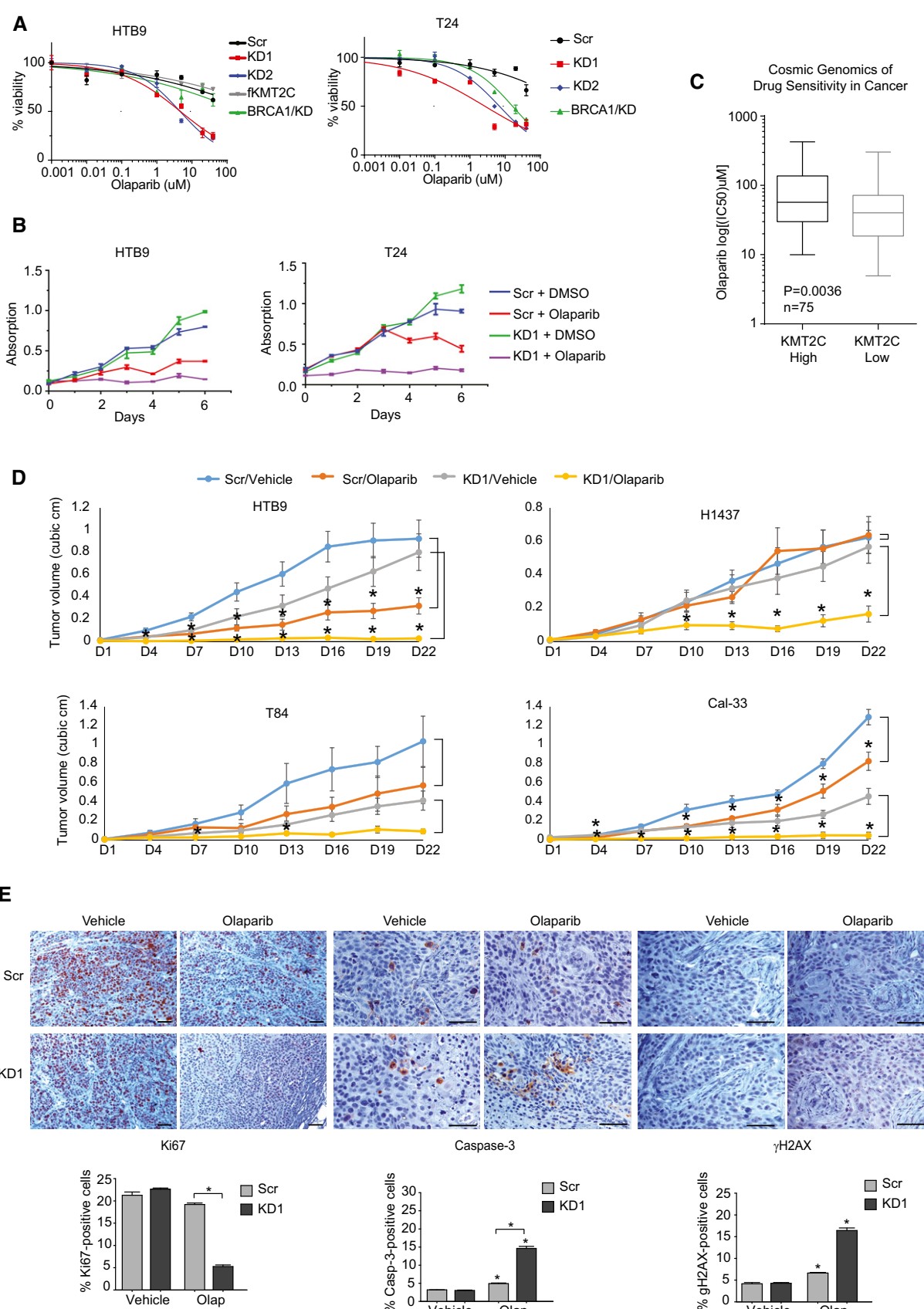

**Figure 8.**

**Figure 8. KMT2C loss leads to PARP1/2 dependence *in vitro* and *in vivo*.**

A MTT assays on untreated control (Scr) and KMT2C/KD cells. BRCA1/KD cells are used as controls. Values represent mean $\pm$ SEM from three experiments.

B MTT assays with IR-treated control (Scr) and KMT2C/KD1 cells treated with 15 μM olaparib. Values represent mean $\pm$ SEM from three experiments.

C Boxplot indicating olaparib IC50 of BLCA, HNSCC, COAD, and NSCLC cell lines from publicly available data (https://www.cancerrxgene.org/). KMT2C$^{high}$ and KMT2C$^{low}$ indicate that KMT2C expression of the cell line is at the top and bottom 50% of the cohort, respectively (data obtained from cbioportal.org; Cancer Cell Line Encyclopedia). Mann–Whitney *U*-test was used.

D Tumor volume obtained from xenografts of control and KMT2C/KD1 cells treated with vehicle or olaparib. The number of mice analyzed for each cohort and raw measurements are provided in Appendix Table S3. The following tumor weight averages (in grams) were obtained $\pm$SEM for vehicle and olaparib, respectively: HTB9/Scr, 0.791 $\pm$ 0.155 and 0.468 $\pm$ 0.097; HTB9/KD1, 0.862 $\pm$ 0.156 and 0.072 $\pm$ 0.023; T84/Scr, 1.032 $\pm$ 0.217 and 0.413 $\pm$ 0.097; T84/KD1, 0.562 $\pm$ 0.159 and 0.105 $\pm$ 0.032; H1437/Scr, 0.661 $\pm$ 0.133 and 0.780 $\pm$ 0.133; H1437/KD1, 0.723 $\pm$ 0.099 and 0.363 $\pm$ 0.108; Cal-33/Scr, 0.439 $\pm$ 0.051 and 0.301 $\pm$ 0.029; Cal-33/KD1, 0.584 $\pm$ 0.224 and 0.05 $\pm$ 0.016. Statistically significant pairwise comparison with respective vehicle for each day is indicated with star. Mann–Whitney *U*-test was used. * designates *P*-value < 0.05. All statistical values including those between Scr and KD1 cells are provided in Appendix Table S4.

E Immunohistochemistry with the indicated antibodies on tumor sections from control (Scr) and KMT2C/KD1 HTB9 cells grown subcutaneously in NOD/SCID mice which were treated with vehicle or olaparib. Statistically significant pairwise comparison with respective vehicle is indicated with stars on top of each column. All other statistically significant comparisons are indicated with squared brackets connecting pairs under comparison. In microscopy images, scale bars indicate 50 μm. In bargraphs, values correspond to mean $\pm$ SEM from n = 3 experiments. Student's *t*-test was used for the analysis. * designates *P*-value < 0.05.

subclones of lung adenocarcinomas during tumor evolution [66] and in metastatic breast cancer subclones [67].

Functionally, KMT2C and the related KMT2D protein direct H3K4 monomethylation, which poises enhancers for activation and transcription factor binding, thus regulating the transcription of neighbor genes [68,69]. In addition to its catalytic role in H3K4me1 deposition, KMT2C interacts with the histone acetyltransferase complex CBP/p300 and the H3K27 demethylase UTX to promote H3K27 acetylation and enhancer activation [68]. The precise control of transcriptional networks through enhancers is important for the tissue-specific expression pattern of developmental genes and plays a crucial role in establishing and maintaining cell fate and identity [70]. Recent studies, however, have shown that alterations in the enhancer epigenetic landscape also correlate with tumorigenesis [71–73]. The reduced expression of KMT2C in bladder epithelial tumor cells leads to a substantial loss of H3K27 acetylation in a subset of active enhancers that control expression of genes involved in focal adhesion, adherens junctions, migration, epithelial cell development, and differentiation. In a recent study by Lin-Shiao et al [34], loss of KMT2D activity by shRNA silencing in primary neonatal human epidermal keratinocytes (NHEKs) and spontaneously immortalized human epidermal keratinocytes (HaCaTs) revealed an important regulatory role of KMT2D in enhancer activity of genes involved in the same processes. This may imply that KMT2C and KMT2D proteins exert coordinated and synergistic functions in enhancer elements and their loss during carcinogenesis deregulates cell adhesion and signaling with profound effects to tumor progression and invasion. Previous reports have shown that concomitant loss of KMT2C and the tumor suppressor protein TP53 expedites tumor formation in mice [13], implying a preferential collaboration between the two. Our own meta-analysis of the publicly available TCGA RNA-seq and mutation data failed to substantiate any consistent correlation between *TP53* mutation status and *KMT2C* expression levels or mutation status (Appendix Fig S5A and B). Moreover, a similar meta-analysis on cancer cell lines also failed to identify any correlation between *TP53* mutation status and *KMT2C* expression (Appendix Fig S5C). Whether the reported connection between KMT2C and TP53 is species-specific or tissue-specific and whether a more universal connection between the proteins exists are questions that would require further experiments to be addressed.

Upon KMT2C reduction, profound gene expression changes are observed. Several genes involved in DDR and DNA repair are downregulated, seemingly due to loss of KMT2C binding on their proximal promoters. Expression downregulation in these cases is associated with reduction in H3K4me3 levels. Although other H3K4 methyltransferases have been found to regulate promoter activity [74,75], this is the first time that KMT2C is found upon promoter regions and implicated in transcription activation of genes, including those encoding DDR and DNA repair proteins. Whether transcription factors mediating oncogenic programs in cancer cells, such as ELK1 downstream of the RAS/MAPK cascade, are responsible for KMT2C recruitment onto promoter regions is a hypothesis that warrants further investigation. It is relevant in this respect that the implication of KMT2C in transcriptional regulation of these genes is confirmed in published TCGA datasets.

Cells with reduced KMT2C levels behave as HR-deficient despite the fact that BRCA1/2 proteins and other HR components are not mutated. HR deficiency as a result of epigenetic regulation of BRCA1/2 expression levels has also been described (reviewed by Konstantinopoulos *et al* [76]). On the other hand, the alt-EJ pathway assumes a critical role, potentially due to HR deficiency. This explains the increased sensitivity of KMT2C/KD cells to PARP1/2 inhibition and offers a promising treatment alternative for KMT2C$^{low}$ cases. Because PARP1/2 participates in the repair of single-strand breaks (SSB; also induced during DNA replication), we cannot exclude the possibility that the increased sensitivity of KMT2C/KD cells to PARPi is also due to unrepaired SSBs which contribute to excess DNA damage. Interestingly, synthetic lethality between PARP1/2 inhibition and BRCA1/2 loss has been solidly established and already exploited in the clinic [77] while a similar link has been established in preclinical models lacking ATM [78]. Our results will hopefully trigger further studies aiming to investigate the relationship between the epigenetic landscape and DNA damage response.

## Materials and Methods

### Human specimens

Bladder tissue specimens were obtained from 138 patients diagnosed with primary urothelial carcinoma of the bladder. Bladder cancer patients underwent transurethral resection of bladder tumors (TURBT) for non-muscle-invasive bladder cancer (Ta, T1) or radical cystectomy (RC) for muscle-invasive bladder cancer (T2–T4). All

 

patients were treated at "Laiko" General Hospital, Athens, Greece (Appendix Tables S1 and S2). Whenever feasible, normal adjacent tissue specimens from 104 of the same patients were included in the study as reference samples, following pathologist's evaluation for absence of CIS and dysplasia. Tissue samples (tumors and normal adjacent specimens) were sectioned into two mirror-image specimens, one of which was submitted to pathologist's evaluation, while the other one was immediately frozen in liquid nitrogen and stored at −80°C until further processing. All patients were diagnosed with urothelial carcinoma on the basis of histopathological criteria, and none of the patients had received any kind of neoadjuvant therapy prior to surgery. No inclusion or exclusion criteria were used other than tissue quality after thawing. Our study was performed according to the ethical standards of the 1975 Declaration of Helsinki, as revised in 2008, and was approved by the ethics committee of "Laiko" General Hospital. Informed consent was obtained from all the participating patients.

## Sanger sequencing

Total RNA from bladder tissue from a previously described cohort [22] was reverse transcribed with random primers and used in PCRs to obtain overlapping amplicons 600–800 bp that cover the portion of *KMT2C* coding region of interest. The oligonucleotide sequences used are provided in Appendix Table S5. PCR fragments were sequenced in both strands with standard Sanger sequencing procedures.

## Cell culture

All cell lines were originally purchased from ATCC. Cells were cultured in Dulbecco's Modified Eagle's Medium (Sigma-Aldrich, cat. D6429) supplemented with 10% heat-inactivated fetal bovine serum (Biosera, cat. FB-1001/500) and penicillin (100 units/ml)/ streptomycin (100 μg/ml; Thermo Fischer Scientific, cat. 15140122) at 37°C with 5% $CO_2$, with the exception of H1437 and H1792 which were cultured in RPMI-1640 medium (Sigma-Aldrich, cat. R8758).

## Lentivirus production, infection, and shRNA knockdown

Scramble, anti-KMT2C, and anti-BRCA1 short hairpin RNA-producing DNA sequences were cloned in PLKO.1-puro-IRES-gfp plasmids. To produce replication-incompetent lentivirus, 293T cells were co-transfected with either Lenti-Scr-GFP or Lenti-shKMT2C-GFP constructs in combination with the pMD2.G and psPAX2 plasmids (Addgene, cat. #12259 and #12260) using the CaCl2 precipitation method. Twelve hours later, growth medium was replenished. Viral supernatants were harvested 36 and 70 h post-transfection. Cell lines were infected overnight with filtered viral supernatants. Three days post-infection, cells were selected with 5–10 μg/ml puromycin (Sigma-Aldrich, cat. P8833) over a period of 7 days. TRCN0000008744 and TRCN000000743 (Sigma-Aldrich) shRNA clones against KMT2C were used for generation of KMT2C/KD1 and KMT2C/KD2 cells, respectively. TRCN0000039834 (Sigma-Aldrich) shRNA clone against BRCA1 was used for generation of BRCA1/KD cells. Scramble, anti-DNA ligase III (TRCN0000048502), and anti-ligase IV (TRCN0000009847) short hairpin RNA-producing DNA sequences (Sigma-Aldrich) were cloned in PLKO.1-blast plasmid

(Addgene: #26655). Lentiviral supernatants generated using these plasmids were used to infect HTB9 and T24 KMT2C/KD1 cell lines. Three days post-infection, cells were selected with 10 μg/ml blasticidin over a period of 12 days for the generation of KMT2C/KD1 cell lines with DNA ligase III or IV knockdown.

## ChIP-seq preparation and analysis

Chromatin was prepared from Scr control and KMT2C/KD HTB9 cells with the SimpleChIP® Enzymatic Chromatin IP Kit (Cell Signaling, cat. 9003) according to the manufacturer's instructions. Libraries were prepared in Greek Genome Center (GGC) Biomedical Research Foundation of Academy of Athens (BRFAA) as previously described [79] and sequenced on the Illumina platform. Single-end 85-bp reads for H3, H3K27ac, H3K4me3, and H3K9ac were generated with the NextSeq500 in the GGC. All ChIP-seq data were aligned to human genome version GCCh37/hg19 with the use of bowtie2 (version 2.1.0) [80] and «−very-sensitive» parameter. Samtools (version 0.1.19) [81] was used for data filtering and file format conversion. MACS (version 1.4.2) algorithm [82] was used for peak calling with H3 ChIP as control. Gene annotation and genomic distribution of the peaks identified by MACs were performed with BEDTools [83], and graph representation (heatmaps) of the tag read density around TSS was performed with seqMiner (version 1.3.3) software [84]. ChIP-seq data have been deposited in the Short Read Archive (SRA) under the BioProject ID: PRJNA508740.

## RNA-seq preparation and analysis

Library preparation for RNA-seq was carried out in the GGC of BRFAA. RNA was isolated from Scr control and KMT2C/KD cells, and RNA-seq libraries were prepared using the TruSeq RNA kit using 1 μg of total RNA. The libraries were constructed according to Illumina's protocols, and equal amounts were mixed and run in the Illumina NextSeq500 in the GGC. Single-end 85-bp reads for three Scr control and three KMT2C/KD samples were generated. RNA-seq raw sequencing data were aligned to human genome version GCCh37/hg19 with the use of TopHat (version 2.0.9) [85] with the use of «−b2-very-sensitive» parameter. Samtools (version 0.1.19) [81] was used for data filtering and file format conversion, while HT-seq count (version 0.6.1p1) algorithm [86] was performed for assigning aligned reads into exons using the following command line «htseq-count –s no –m intersection -nonempty». Finally, differentially expressed genes were identified with the use of the DESeq R package [87] and genes with fold change cutoff 1.5 and *P*-adj ≤ 0.05 were considered to be differentially expressed genes (DEGs). Gene ontology and pathway analysis was performed in the DEGs with the DAVID knowledge base [88] and Ingenuity Pathway Analysis software (IPA). Only pathways and biological processes with *P*-value ≤ 0.05 were considered significantly enriched. RNA-seq data have been deposited in the Short Read Archive (SRA) under the BioProject ID: PRJNA508526.

## CMA

The chromosomal microarray analysis (CMA) was performed with the high-resolution 2 × 400 K G3 CGH+SNP microarray platform (G4842A, Design ID 028081, Agilent Technologies, Santa Clara, CA,

USA). The specific platform features a total of 292,097 oligonucleotide CGH probes covering the whole genome, with a median CGH probe spacing of 7 kb, as well as 118,955 single nucleotide polymorphism (SNP) probes for the detection of copy neutral loss of heterozygosity (CN-LOH), resulting in a resolution of 5–10 Mb for CN-LOH. The wet-lab protocol was according to the manufacturer's instructions (Agilent Oligonucleotide Array-Based CGH for Genomic DNA Analysis) and consisted of enzymatic digestion of genomic DNA in parallel with a sex-matched reference DNA (Agilent Technologies) with restriction enzymes AluI and RsaI, followed by differential labeling with Cy3 and Cy5 fluorescent dyes for sample and reference, respectively. Following purification, the combined labeled DNA samples were applied to the microarray (hybridization for 40 h at 67°C), washed, and scanned at 3 micron resolution on the Agilent High-Resolution Microarray Scanner (G2505C, Agilent Technologies). The images were extracted and analyzed using the Agilent Feature Extraction software and the CytoGenomics v.3.0 software suite. The ADM-1 aberration detection algorithm was utilized, and the minimum number of probes required for a call was set to 4.

### Real-time qPCR

For human tissue samples, total RNA was isolated, following the pulverization of 40–100 mg of bladder tissue specimens, with the use of TRI reagent (Molecular Research Center, Inc., Cincinnati, OH, USA) and reverse transcribed with MMLV reverse transcriptase (Invitrogen, Carlsbad, CA, USA) using oligo-dT primers. For cell lines, total RNA was isolated with the use of TRI reagent and reverse transcribed with PrimeScript™ RT reagent Kit (Takara, RR037A) using oligo-dT and random primers. Quantitative PCR was performed in the 7500 Real-Time PCR System using the sequence detection software (Applied Biosystems, Carlsbad, CA, USA). The 10 µl reaction mixture consists of Kapa SYBR Fast Universal 2× qPCR Master Mix (Kapa Biosystems, Inc., Woburn, MA, USA). Melting curve analysis was performed following the amplification in order to distinguish specific reaction products from non-specific ones or primer dimers. Gene expression analysis was carried out using the $2^{-\Delta\Delta C_T}$ relative quantification method [89]. Duplicate or triplicate reactions were performed for each tested sample, and the average $C_T$ was calculated for the quantification analysis. *HPRT1* was used as an endogenous reference control. Oligonucleotide sequences are provided in Appendix Table S6.

### Immunofluorescence experiments

Cells were plated on poly-L-lysine (Sigma, cat. P1274)-coated coverslips. Cells were fixed by 10-min incubation in 4% paraformaldehyde (Alfa Aesar, 30525-89-4) at room temperature, permeabilized for 4 min in 1× PBS/0.5% Triton X-100, washed with PBS, and blocked in 1% bovine serum albumin (Applichem, cat. A1391,0100) and 10% fetal bovine serum in PBS. Cells were incubated with primary antibody (Appendix Table S7) overnight at 4°C, followed by incubation with a fluorescent secondary antibody for 1 h at room temperature as previously described [90]. Antibody solutions were made in PBS with 1% bovine serum albumin. Coverslips were mounted on glass slides using VECTASHIELD Antifade Mounting Medium with 49,6-diamidino-2-phenylindole (DAPI) for DNA

staining (Vektor, cat. H-1200). For DNA repair experiments, cells were treated with 2 µM cisplatin for 6 h. For tissue stainings, tumors were fixed in 4% formaldehyde at 4°C, thoroughly washed in PBS, placed in 30% sucrose overnight, and frozen in optimal cutting temperature (OCT) compound (Tissue Tek, Sakura). Frozen 10 µm sections were obtained using a Leica (CM1950) cryostat. Sectioned tissues were washed three times in PBS, blocked for 1 h, and incubated with primary and secondary antibodies as described. Image processing and foci counts were performed using ImageJ.

### Cytogenetics and FISH

Standard procedures were used for chromosome preparation and staining [91]. Briefly, cells were treated with 10 µg/ml Colcemid™ (Thermo Fisher Scientific, catalog No. 15210040) for 1 h, harvested, treated with 75 mM KCl for 20 min, fixed in methanol/glacial acetic acid (3:1, v/v), and processed for cytogenetic analysis. Imaging and karyotyping were performed via microscopy and computer imaging techniques. At least 30 metaphases per cell line were karyotyped. Karyotypes were analyzed according to the International System for Human Cytogenetic Nomenclature (ISCN) 2013.

### MTT cell viability assays

For cell viability assays, cells were plated at $6–10 \times 10^3$ per well in 48-well plates and incubated with complete Dulbecco's Modified Eagle's Medium (DMEM) or RPMI-1640 medium containing different concentrations of olaparib as indicated in the respective figure legend. Assays were performed using the standard MTT colorimetric assay (Sigma, cat. M5655) according to the manufacturer's instructions. Measurements were analyzed using GraphPad Prism v6.

### Soft-agar clonogenic assays

Basal anchorage-independent growth inhibition of olaparib was assessed by a double-layer soft-agar assay. Cells ($5 \times 10^4$) were suspended in complete medium containing 0.35% agar and increasing olaparib concentrations, and seeded in triplicate in 24-well plates onto a base layer of complete medium containing 1% agar. Medium was replenished every 3–5 days for 15–20 days, before colony counting. Image processing and colony counts were performed using ImageJ.

### Mice

Male NOD/SCID mice were purchased from the Jackson repository and bred in individually ventilated cages at the Animal House Facility of the Foundation for Biomedical Research Foundation of the Academy of Athens (Athens, Greece) under veterinarian supervision. All procedures for care and treatment of animals were approved by the Institutional Committee on Ethics of Animal Experiments and the Greek Ministry of Agriculture. Cells were injected when mice reached the age of 4–6 weeks. For *in vivo* treatments, cells were injected in the flanks, and when tumors reached a palpable size (2–3 mm in diameter or dimension), mice were randomly assigned to groups. No exclusion criterion was applied. The olaparib therapy group was intraperitoneally administered with either vehicle or olaparib injection (AZD2281, MedChem express, at a dose

of 50 mg/kg in PBS solution containing 12.5% DMSO and 12.5% kolliphor) following the cycling dosing scheme OROOR (O: olaparib or Vehicle, R: Rest) for 21 days. Tumors were dissected and weighed 24 h after the last treatment. Tumor volume measurements were performed every 3 days using a caliper.

### Ionizing radiation experiments

Irradiation was carried out in a GammaCell 220 irradiator (Atomic Energy of Canada Ltd., Ottawa, Canada) at room temperature. For chromosomal fusion events analysis, cells synchronized at the G0/G1 phase by growing the cell cultures to confluency followed by serum deprivation for 48 h (0.1–0.25% serum) [92]. Cells were exposed to ionizing radiation (6 Gy) and incubated at 37°C for 6 h to recover. At 6 h post-irradiation, cells were trypsinized and cultured with fresh medium for 30 h. Subsequently, cells were treated with colcemid for 1 h to arrest dividing cells at metaphase and processed for chromosome preparation and staining as previously described. For drug treatments, cells were exposed to 15 μM olaparib and/or 30 μM SCR7 (MedChem Express, cat. HY-12742) 24 h prior to irradiation until metaphase harvesting. For each experiment, 30 metaphases were scored. Experiments were repeated thrice. Light microscopy was coupled to an image analysis system (MetaSystems, Altlussheim, Germany) to facilitate scoring.

### Protein extraction and Western blot analysis

Cells were lysed in RIPA lysis buffer (150 mM NaCl, 50 mM Tris–HCl pH 8, 1% IGEPAL, 0.5% sodium deoxycholate, 0.1% SDS) that was added with a protease inhibitor cocktail (Complete, Roche). A total protein amount of 20 μg from each samples was denatured at 95°C for 10 min in Laemmli buffer containing β-mercaptoethanol before electrophoresis. The primary antibodies that were used are described in Appendix Table S7.

### DNA fiber assay

Asynchronous cell cultures were treated with 25 μM IdU (Sigma-Aldrich, I7125) for 20 min, washed with PBS, and exposed to 250 μM CldU (Sigma-Aldrich, C6891) for 20 min (−HU) or exposed to 250 μM CldU and 0.2 mM hydroxyurea for 60 min (+HU). After exposure to CldU, cells were washed in PBS and harvested. Cells were then lysed and DNA fibers stretched onto glass slides and fixed as described [93]. Fibers were denatured with 2.5 M HCl for 80 min, washed with PBS, and blocked with 2% BSA in phosphate buffer saline for 30 min. The newly synthesized IdU and CldU tracts were visualized with anti-BrdU antibodies recognizing IdU (1:50 BD Biosciences, 347580) and CldU (1:400, Abcam, ab6326), respectively. Images were taken at 60× magnification using a Leica DM RA2 fluorescence microscope equipped with a Hamamatsu ORCA-Flash 4.0 V2 (sCMOS-Monochrome, 4 Mpixel) camera and analyzed using ImageJ software. Statistical analysis was carried out using GraphPad Prism.

### Sister chromatid exchange

Cells ($1 \times 10^5$) were plated on a 10-cm dish. At 24 h, cells were treated with 5 μM cisplatin. At 3 h post-cisplatin treatment, cells were washed with fresh medium and treated with 5 mg/ml BrdU (Sigma-Aldrich, cat. B5002) for 40 h followed by 0.2 μg/ml colchicine for 3 h. Sister chromatid exchange assays were performed as previously described [94].

### Statistical analysis

In human tissue samples, the normality of the distribution of MLL3/KMT2C expression in bladder tissue specimens was evaluated by Shapiro–Wilk test. The nonparametric Wilcoxon signed-rank test was used to analyze MLL3/KMT2C expression between bladder tumor specimens and matched adjacent normal tissues. Animals were randomly assigned into different groups. Group allocation and outcome assessment were not blinded. In two group comparisons, normality of distribution was determined by D'Agostino & Pearson omnibus normality test and Shapiro–Wilk normality test (paired *t*-test). For non-Gaussian sample distribution or small sample size, Mann–Whitney *U*-test was employed. Sample sizes met the minimum requirements of the respective statistical test used. A value of $P < 0.05$ was considered as significant. Animals which did not develop tumors or did not live through the end of the treatment were excluded. Mann–Whitney *U*-test was also employed for statistical evaluation of chromosome number differences in karyotyping experiments and tumor volumes in mouse xenograft experiments.

Expanded View for this article is available online.

### Acknowledgements

We would like to thank Dr. E. Chavdoula for providing NOD/SCID mice, Dr. M. Roubelaki for help with the cell cycle analysis, Dr. N. Paschalidis for Annexin V FACS analysis and Drs. V. Paraskevopoulou and P. Karakaidos for critically reading the manuscript. This work was supported by a Greek General Secretariat for Research and Technology "Excellence" grant (UTN_1799), a Fondation Santé Grant in Biomedical Sciences, a Worldwide Cancer Research grant (16/1217) and a Horizon 2020 grant (732309) to AK, and a Greek General Secretariat for Research and Technology and the Hellenic Foundation for Research and Innovation (HFRI) grant (472-EpiNotch) to TR.

### Author contributions

AKl and TR conceived the study and designed all experiments. TR, DK, MA, AKo, ZK, and EKo performed all experiments. AP analyzed expression array data. AS and KS provided the library of human tumor samples. KNM and GEP performed the IR and karyotyping experiments. MT and EKa performed and analyzed CMA experiments. AKl wrote the manuscript.

### Conflict of interest

The authors declare that they have no conflict of interest.

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
