## [Review Process File · EMBO Reports]

The lysine-specific methyltransferase KMT2C/MLL3 regulates DNA repair components in cancer

Theodoros Rampias, Dimitris Karagiannis, Margaritis Avgeris, Alexander Polyzos, Antonis Kokkalis, Zoi Kanaki, Evgenia Kousidou, Maria Tzetis, Emmanouil Kanavakis, Konstantinos Stravodimos, Kalliopi N. Manola, Gabriel E. Pantelias, Andreas Scorilas and Apostolos Klinakis

Review timeline:

Submission date:	27 July 2018
Editorial Decision:	1 August 2018
Revision received:	7 November 2018
Editorial Decision:	27 November 2018
Revision received:	17 December 2018
Accepted:	19 December 2018

Editor: Achim Breiling

Transaction Report: This manuscript was transferred to *EMBO reports* following peer review at *The EMBO Journal*

1st Editorial Decision

1 August 2018

Thank you for the transfer of your research manuscript to EMBO reports. I have now read your paper and went through the referee reports from The EMBO Journal (which you will find attached at the end of this message).

All referees acknowledge the potential interest of the findings. Nevertheless, all referees have raised a number of concerns and suggestions to improve the manuscript, or to strengthen the data and the conclusions drawn. As the reports are below, I will not detail them here.

As EMBO reports emphasizes novel functional over detailed mechanistic insight, we will not require to address the points regarding more refined mechanistic details. However, manuscripts accepted by EMBO Reports should communicate robustly documented, major findings of physiological relevance. Therefore, it is important to address the major points raised by referees #1 and #3 experimentally. Further, please also address the minor points made by referees #2 and #3.

Given the constructive referee comments, we would like to invite you to revise your manuscript for EMBO reports with the understanding that all referee concerns must be addressed in the revised manuscript and in a point-by-point response. Acceptance of your manuscript will depend on a positive outcome of a second round of review. It is our policy to allow a single round of revision only and acceptance or rejection of the manuscript will therefore depend on the completeness of your responses included in the next, final version of the manuscript.

Revised manuscripts should be submitted within three months of a request for revision; they will otherwise be treated as new submissions. Please contact us if a 3-months time frame is not sufficient for the revisions so that we can discuss the revisions further.

Please refer to our guidelines for preparing your revised manuscript and the figure panels:
<http://embor.embopress.org/authorguide#manuscriptpreparation>

http://embopress.org/sites/default/files/EMBOPress_Figure_Guidelines_061115.pdf

Supplementary/additional data: The Expanded View format, which will be displayed in the main HTML of the paper in a collapsible format, has replaced the Supplementary information. You can submit up to 5 images as Expanded View. Please follow the nomenclature Figure EV1, Figure EV2 etc. The figure legend for these should be included in the main manuscript document file in a section called Expanded View Figure Legends after the main Figure Legends section. Additional Supplementary material should be supplied as a single pdf labeled Appendix. The Appendix includes a table of content on the first page, all figures and their legends. Please follow the nomenclature Appendix Figure Sx throughout the text and also label the figures according to this nomenclature. For more details please refer to our guide to authors.

Important: All materials and methods should be included in the main manuscript file.

Regarding data quantification and statistics, can you please specify, where applicable, the number "n" for how many independent experiments (biological replicates) were performed, the bars and error bars (e.g. SEM, SD) and the test used to calculate p-values in the respective figure legends. Please provide statistical testing where applicable. See:
<http://embor.embopress.org/authorguide#statisticalanalysis>

Please also follow our guidelines for the use of living organisms, and the respective reporting guidelines: <http://embor.embopress.org/authorguide#livingorganisms>

We now strongly encourage the publication of original source data with the aim of making primary data more accessible and transparent to the reader. The source data will be published in a separate source data file online along with the accepted manuscript and will be linked to the relevant figure. If you would like to use this opportunity, please submit the source data (for example scans of entire gels or blots, data points of graphs in an excel sheet, additional images, etc.) of your key experiments together with the revised manuscript. Please include size markers for scans of entire gels, label the scans with figure and panel number. Please send one PDF file per figure.

Finally, please format the references according to our journal style. See:
<http://embor.embopress.org/authorguide#referencesformat>

- a complete author checklist, which you can download from our author guidelines (<http://embor.embopress.org/authorguide#revision>). Please insert page numbers in the checklist to indicate where the requested information can be found.
- a letter detailing your responses to the referee comments in Word format (.doc)
- a Microsoft Word file (.doc) of the revised manuscript text
- editable TIFF or EPS-formatted single figure files in high resolution (for main figures and EV figures)

I look forward to seeing a revised version of your manuscript when it is ready. Please let me know if you have questions or comments regarding the revision.

Referee #1:

In this study, Rampias et al., describe the role of KMT2C in maintenance of genome stability through transcriptional regulation of DNA repair genes especially those involved in homologous recombination. It is a solid study and their findings are interesting and well documented. The study includes work in human patients' samples, mechanistic insights in cell lines and mouse xenografts. Even if the role of KMT2C in genome stability is indirect through transcriptional regulation it is still interesting and clinically relevant.

For strengthening the study, the authors need to clarify the specificity of the role of KMT2C in genomic instability in certain cancers like bladder cancers by showing that in non epithelia or

myeloid driven cancers the effects of downregulation of KMT2C are not the same.

Referee #2:

This study focuses on the lysine-specific methyltransferase 2C (KMT2C) that appears to act as a tumor suppressor in some cancers. Data is presented showing that the KMT2C gene function is frequently reduced in bladder cancers either through mutation or promoter hypermethylation. Given its function as an epigenetic regulator, knockdown of KMT2C resulted in altered expression of a large number of genes, including genes involved in the repair of DNA double strand breaks by homologous recombination. Interestingly, transcription regulation by KMT2C occurred not only at enhancers but also at regions close to transcriptional start sites.

Expression of genes involved in a number of pathways implicated in cancer initiation and progression were altered in KMT2C deficient cells with pathways involved in the DNA damage response and recombination repair prominently represented. Further studies showed that, as expected, this resulted in increased genome instability and sensitivity to PARP Inhibitors. These findings have important translational implications as PARP inhibitors have shown promising activity in tumors with defined genetic defects in recombinational repair and there is emerging evidence that this type of defect is relatively common in sporadic cancers. Thus, the identification of alterations in KMT2C as a biomarker of defective recombinational repair has the potential to identify patient populations whose disease is likely to respond to PARP inhibitors.

That being said, the advance in mechanistic understanding is limited and so the study is probably better suited to a more specialized journal

Minor comments:

- Figure legends are too brief and lacking in detail
 - In some figures (for example Fig. 8), the effect of knockdown on protein steady state levels is not shown
- Effect of DNA damaging agents and/or PARP inhibitors on colony formation should be shown in addition to MTT assays
- There are contradictory reports as to the activity and specificity of SCR7. The authors should consider using DNA PK inhibitors to block NHEJ

Referee #3:

In this manuscript, Rampias and colleagues investigate the tumor suppressor role of KMT2C in bladder cancer tumorigenesis. In the manuscript, they demonstrate KMT2C loss and its consequential effects at enhancer and promoter regions of genes. Amongst the genes that are down-regulated by KMT2C at promoters are genes involved in the DNA Damage Response pathway as revealed by RNA-seq and GO analysis. In order to pinpoint the mechanism, the authors create a KMT2C knockdown cell line to reveal its effects in genome stability. The data from DNA damage experiments reveal that KMT2C reduction causes a downregulation of genes involved in the DNA Damage Repair pathway, and increases the level of damaged DNA causing it to rely heavily upon the alternative- End Joining (alt-EJ) pathway, a highly error prone DNA repair pathway, which leads to genome instability. To hijack the perceived vulnerability of KMT2C loss in bladder cancers and its dependence on alt-EJ, the authors use Olaparib, a PARP inhibitor to reveal a synergistic effect between KMT2C loss and PARP inhibition. Cell viability assays and in vivo experimentation using tumor volume reveal that KMT2C knockdown and Olaparib treatment has a synergistic effect. Taken together, the data reveal a promising treatment option for bladder cancer patients with KMT2C low expression and PARP inhibition.

The manuscript is interesting, novel and potentially clinically important. I suggest a few points below that could further strengthen it and increase its significance.

Major points:

- In Dawkins et al., Cancer Research 2017, the authors showed that KMT2C inhibition resulted in

downregulation of several genes, including DNA repair related genes. However, they found that KMT2C inhibition reduced cell viability, which seems to contradict the current finding in this manuscript. The author mentioned there are no consequences on cell viability in BC cells, however no data was shown to demonstrate this. Can the authors perform Annexin-V staining to demonstrate that KMT2C does not affect viability in their system?

- To demonstrate the tumor suppressor role of KMT2C, does overexpression or restoration of KMT2C in KMT2C mutated or downregulated tumors (or cell lines) decrease cell viability?
- In Figure 3E rescue experiments, have the authors performed domain mutations (those found commonly mutation in the PHD1-3 domain) to test if KMT2C can rescue? Since PHD1-3 mutations also result in missense mutations that may or may not affect the total stability of KMT2C.
- In Figure 8D, please overlap all the points for SCR and KD1 for each cell line xenograft. It seems that there's only a slight decrease in terms of tumor size when comparing scr+Olaparib to KD1 +Olaparib. Have the authors performed a Kaplan Meier survival analysis to see if there's an overall survival advantage?
- In the cited references, Lee, Kim et al., 2009 demonstrated that the p53 mutational status plays a crucial role in KMT2C loss-of-function disease progression. Therefore, does KMT2C mutations or downregulation correlated with p53 mutations? This is potentially important, as it can influence the synergy with PARP inhibitors.
- In Figure 1A, the authors show many KMT2C missense/frameshift mutations, however there is not much focus on these specific mutations later in the paper. Additionally, many of these mutations are missense and the authors predicted to have damaging effects, however it has not been shown experimentally and will need to be validated further therefore it's inconclusive to state these are loss-of-function mutations.

Minor points:

- In Figure 1C, immunofluorescence provided more information on the localization and therefore we recommend western blot analysis to look at protein levels of KMT2C.
- In Figure 1D, COAD boxplot, there is no label on the x-axis. Is it comparing Normal vs. Tumor?
- Figure 2C, scr control is recommended to compare the other two conditions (KD1&fkKMT2C) and p-values are needed for error bars.
- In Figure 2F, what's the significance of the genes stated in this panel and how does it contribute to the paper? It is unclear how these genes play a role in BC or to the paper. Can the authors comment on this.
- Figure 2G, We do see a decrease in H3K27ac at enhancers but there's also a decrease in binding at the promoter ITGB1, can the authors explain this result?
- In the manuscript text, the authors mention controls, however the term is vague and can be specified (empty vector control, shRNA control, etc...)
- In Figure 2, how does the expression of KMT2C in HTB9 and T24 compared to other cancer cell lines? Is it considered highly expressed? Why were these cell lines chosen?
- In Figure 2, the authors showed knockdown of KMT2C in two independent lines (HTB9 and T24). However, in the rest of figure 2 RNA-seq/ChIP-seq analysis, was this done in comparing both cell lines? Or just HTB9? To demonstrate the broad transcriptional network of KMT2C, another independent cell line is required.
- The results need p-values in Figure 3D.
- In Figure 4B, please label the type of cancers associated with these cell lines. Also among the other

cell lines, H1792 cells did not decrease. Can the authors comment? Also are all these cell lines low in KMT2C expression?

1st Revision - authors' response

7 November 2018

Referee #1:

In this study, Rampias et al., describe the role of KMT2C in maintenance of genome stability through transcriptional regulation of DNA repair genes especially those involved in homologous recombination. It is a solid study and their findings are interesting and well documented. The study includes work in human patients' samples, mechanistic insights in cell lines and mouse xenografts. Even if the role of KMT2C in genome stability is indirect through transcriptional regulation it is still interesting and clinically relevant.

For strengthening the study, the authors need to clarify the specificity of the role of KMT2C in genomic instability in certain cancers like bladder cancers by showing that in non-epithelia or myeloid driven cancers the effects of downregulation of KMT2C are not the same.

This is a good point raised by the reviewer. We have now included both expression and mutation TCGA data from hemopoietic, nervous system and soft tissues indicating that KMT2C expression levels are higher in the tumor compared to healthy tissue (Supplementary Figure 1). This suggests that KMT2C downregulation is not favorable in non-epithelial or myeloid driven cancer types. Interestingly, the expression levels of KMT2C correlate with those of ATR, ATM, BRCA1, BRCA2 DNA repair proteins in all cancer types tested, in agreement with our findings (Incorporated as Supplementary Figure 5). This implies that the transcriptional regulation of genes encoding DDR/DNA repair components by KMT2C is more global. Whether higher KMT2C expression in these tumors affects DNA repair is an interesting question that would need to be addressed in a different piece of work.

Referee #2:

This study focuses on the lysine-specific methyltransferase 2C (KMT2C) that appears to act as a tumor suppressor in some cancers. Data is presented showing that the KMT2C gene function is frequently reduced in bladder cancers either through mutation or promoter hypermethylation. Given its function as an epigenetic regulator, knockdown of KMT2C resulted in altered expression of a large number of genes, including genes involved in the repair of DNA double strand breaks by homologous recombination. Interestingly, transcription regulation by KMT2C occurred not only at enhancers but also at regions close to transcriptional start sites.

Expression of genes involved in a number of pathways implicated in cancer initiation and progression were altered in KMT2C deficient cells with pathways involved in the DNA damage response and recombination repair prominently represented. Further studies showed that, as expected, this resulted in increased genome instability and sensitivity to PARP Inhibitors. These findings have important translational implications as PARP inhibitors have shown promising activity in tumors with defined genetic defects in recombinational repair and there is emerging evidence that this type of defect is relatively common in sporadic cancers. Thus, the identification of alterations in KMT2C as a biomarker of defective recombinational repair has the potential to identify patient populations whose disease is likely to respond to PARP inhibitors. That being said, the advance in mechanistic understanding is limited and so the study is probably better suited to a more specialized journal.

We thank the reviewer for his/her positive comments.

Minor comments;

Figure legends are too brief and lacking in detail

We have tried to make figure legends more elaborative.

In some figures (for example Fig. 8), the effect of knockdown on protein steady state levels is not shown

It is not clear what the reviewer is referring to. The protein levels of KMT2C have been assessed with WB in Figure 2 for both the HTB9 and T24 cell lines with which the majority of experiments has been performed. For BRCA1, we have included the remaining protein levels in Supplementary Figure 6A. Ligase III and IV western blots can be found in Supplementary Figure 8.

Effect of DNA damaging agents and/or PARP inhibitors on colony formation should be shown in addition to MTT assays

Following the reviewer's advice we have performed clonogenic assays. The data obtained do not deviate from those from MTT assays. These data are presented as Figure S9 in the revised manuscript.

There are contradictory reports as to the activity and specificity of SCR7. The authors should consider using DNA PK inhibitors to block NHEJ

This is an absolutely valid point. Instead of using DNA-PK inhibitors, we have used genetic means to knockdown Ligase IV (and independently Ligase III) levels to dissect the roles of canonical and alternative nonhomologous end joining. We performed cytogenetics analysis to measure unrepaired damage after irradiation and the obtained results do not deviate from those acquired by SCR7 and olaparib treatment (Supplementary Figure S8).

Referee #3:

In this manuscript, Rampias and colleagues investigate the tumor suppressor role of KMT2C in bladder cancer tumorigenesis. In the manuscript, they demonstrate KMT2C loss and its consequential effects at enhancer and promoter regions of genes. Amongst the genes that are down-regulated by KMT2C at promoters are genes involved in the DNA Damage Response pathway as revealed by RNA-seq and GO analysis. In order to pinpoint the mechanism, the authors create a KMT2C knockdown cell line to reveal its effects in genome stability. The data from DNA damage experiments reveal that KMT2C reduction causes a downregulation of genes involved in the DNA Damage Repair pathway, and increases the level of damaged DNA causing it to rely heavily upon the alternative- End Joining (alt-EJ) pathway, a highly error prone DNA repair pathway, which leads to genome instability. To hijack the perceived vulnerability of KMT2C loss in bladder cancers and its dependence on alt-EJ, the authors use Olaparib, a PARP inhibitor to reveal a synergistic effect between KMT2C loss and PARP inhibition. Cell viability assays and in vivo experimentation using tumor volume reveal that KMT2C knockdown and Olaparib treatment has a synergistic effect. Taken together, the data reveal a promising treatment option for bladder cancer patients with KMT2C low expression and PARP inhibition.

The manuscript is interesting, novel and potentially clinically important. I suggest a few points below that could further strengthen it and increase its significance.

Major points:

- *In Dawkins et al., Cancer Research 2017, the authors showed that KMT2C inhibition resulted in downregulation of several genes, including DNA repair related genes. However, they found that KMT2C inhibition reduced cell viability, which seems to contradict the current finding in this manuscript. The author mentioned there are no consequences on cell viability in BC cells, however no data was shown to demonstrate this. Can the authors perform Annexin-V staining to demonstrate that KMT2C does not affect viability in their system?*

This is a correct point. We have now included a new Figure (Supplementary Figure 3 in the revised manuscript) presenting cell cycle analyses, annexin V staining and FACS analysis, as well as MTT assays supporting our claim.

- *To demonstrate the tumor suppressor role of KMT2C, does overexpression or restoration of KMT2C in KMT2C mutated or downregulated tumors (or cell lines) decrease cell viability?*

The tumor suppressor role of KMT2C has been addressed in mouse models and involves both the urinary bladder, as the reviewer himself mentions in a following comment. Mice and/or primary cells are, in our opinion, more suitable models to indicate tumor suppressor roles for any protein. This has been unequivocally proven for KMT2C in mice, while human sequencing data support such a role. Tumor cell lines are more appropriate in studies such as this one aiming to indicate the

role of KMT2C in transcription regulation in already tumor cells. Besides, from the cell lines at hand, to our knowledge, only the BC cell line RT112 carries a missense mutation, the contribution of which in tumor development is not known. Similarly, we cannot be sure whether the presumptive reduction of KMT2C levels in our cell lines have been causally involved in tumor development. Thus, complementing the function of KMT2C in a series of cells would not necessarily yield conclusive evidence as to its tumor suppressor role.

• In Figure 3E rescue experiments, have the authors performed domain mutations (those found commonly mutation in the PHD1-3 domain) to test if KMT2C can rescue? Since PHD1-3 mutations also result in missense mutations that may or may not affect the total stability of KMT2C.

This would be out of the scope of this manuscript that focuses on the expression levels of KMT2C in cancer tissue. We assume that the reviewer refers to Figure 3F, in which re-expression of wild-type KMT2C rescues the expression of DDR genes. In a recent manuscript (Wang L, Zhao Z, et al (2018) Resetting the epigenetic balance of Polycomb and COMPASS function at enhancers for cancer therapy. Nat Med 24: 758-769), it is clearly shown that missense mutations in the PHD domain affect the ability of the protein to interact with BAP1 and thus get recruited onto chromatin. This is now discussed in the manuscript in Figure 1 (middle of page 4).

• In Figure 8D, please overlap all the points for SCR and KD1 for each cell line xenograft. It seems that there's only a slight decrease in terms of tumor size when comparing scr+Olaparib to KD1 +Olaparib. Have the authors performed a Kaplan Meier survival analysis to see if there's an overall survival advantage?

Following the reviewer's advice, we have merged the plots. We do not exactly understand what the reviewer means by "slight decrease". Olaparib has a variable effect on Scr cells varying from significant in HTB9 cells to negligible in H1437. In KD1 cells, independently of their growth rate under vehicle, the effect of Olaparib is detrimental. In fact, these cells practically fail to grow at all. A Kaplan-Meier would be impossible in this case. Because tumors grow subcutaneously, they do not affect vital organs. Thus, mice survive tumor masses almost comparable to their actual size. All mice need to be sacrificed when tumors reach a size of about 1.5 to 2 cm due to ethical issues.

• In the cited references, Lee, Kim et al., 2009 demonstrated that the p53 mutational status plays a crucial role in KMT2C loss-of-function disease progression. Therefore, does KMT2C mutations or downregulation correlated with p53 mutations? This is potentially important, as it can influence the synergy with PARP inhibitors.

This is an interesting point. To this direction, we performed an extensive meta-analysis of tumor and cell line expression and mutational data in order to investigate a possible relationship between KMT2C and TP53. As Supplementary Figure S10 indicates, KMT2C downregulation does not correlate either TP53 expression nor mutation status. Neither do KMT2C mutations coexist with TP53 mutations. This is now discussed in the Discussion section of the revised manuscript (middle of p. 15).

• In Figure 1A, the authors show many KMT2C missense/frameshift mutations, however there is not much focus on these specific mutations later in the paper. Additionally, many of these mutations are missense and the authors predicted to have damaging effects, however it has not been shown experimentally and will need to be validated further therefore it's inconclusive to state these are loss-of-function mutations.

Following the 2014 TCGA publication on bladder cancer which identified KMT2C mutations at high frequency, we considered that a major question on whether these mutations appear early in the process of tumorigenesis remained open. This is due to the fact that the TCGA cohort is composed of exclusively high grade/late stage tumors. Thus, we decided that this should be settled. We performed the mutation analysis using a balanced cohort of clinical samples (superficial and invasive tumors are represented with the same frequency they are diagnosed in the clinic) and included these data in this manuscript, although its major focus is indeed the transcriptional downregulation of KMT2C. In the meantime, a recent paper (2017, by the Margaret Knowles group) also reported that KMT2C mutations are equally distributed between superficial and muscle invasive bladder tumors.

With respect to characterizing these mutations as loss-of-function, our initial assessment was based on the Polyphen-2 algorithm, which has proven credible in our hands in previous studies. In the meantime, a paper by Shilatifard and co-workers showed that missense mutations within the PHD domains are commonly loss-of-function because they disrupt KMT2C interaction with BAP1, and

thus KMT2X recruitment to gene enhancers (Wang L, Zhao Z, et al (2018) Resetting the epigenetic balance of Polycomb and COMPASS function at enhancers for cancer therapy. Nat Med 24: 758-769). This is discussed in the middle of page 4.

In general, we avoid characterizing these mutations as loss-of-function and refer to them as presumably or likely. Functionally characterizing these mutations would be definitely out of the scope of this manuscript.

Minor points:

• *In Figure 1C, immunofluorescence provided more information on the localization and therefore we recommend western blot analysis to look at protein levels of KMT2C.*

Following the reviewer's advice, we performed western blot on the same samples. As the revised Figure 1 indicates, indeed there is a downregulation of KMT2C at the protein level, matching the qPCR data. We apologize for the average quality of the film, however, it is extremely difficult to obtain better results from clinical samples for a protein of 500 KDa.

• *In Figure 1D, COAD boxplot, there is no label on the x-axis. Is it comparing Normal vs. Tumor?*

We apologize for the confusion. Yes, it is comparing normal vs. tumor and the Y axis is the ratio (log[normal/tumor]). The reason for this is that for COAD, healthy samples are analyzed only with Affymetrix microarrays while tumor samples have been analyzed with both Affymetrix and illumina RNA-seq. Thus, we used only the microarray data for the comparison. This has been clearly stated in the figure legend in the revised manuscript.

• *Figure 2C, scr control is recommended to compare the other two conditions (KD1&fMT2C) and p-values are needed for error bars.*

Scr control is part of the analysis. What is shown in the Y axis is the ratio of KD1/Scr and fKMT2C/Scr. This has been more clearly stated in the legend.

• *In Figure 2F, what's the significance of the genes stated in this panel and how does it contribute to the paper? It is unclear how these genes play a role in BC or to the paper. Can the authors comment on this.*

The role of KMT2C in enhancer H3K4 monomethylation is well established. However, our ChIP-Seq data as well as ChIP-Seq data from other groups (Wang L, Zhao Z, et al (2018) Resetting the epigenetic balance of Polycomb and COMPASS function at enhancers for cancer therapy. Nat Med 24: 758-769) show that a high percentage of KMT2C binding is distributed among promoter regions. In this direction we performed a separate analysis of KMT2C peaks in enhancer and promoter regions in order to measure the degree of overlap between genes with either KMT2C enhancer or promoter binding. Interestingly, our analysis revealed a small percentage of genes with dual binding, supporting a discrete functional role for KMT2C in transcriptional regulation via promoter binding. This is highlighted by the fact that the identified DNA binding motifs corresponding to enhancer and promoter binding are totally different. KMT2C enhancer binding is mostly found in genes which are involved in epithelial integrity, epithelial differentiation and cell-cell interactions whereas, KMT2C promoter binding is detected mostly in genes implicated in genome integrity and homologous recombination DNA repair. While the latter group of genes is the core of the paper, there is extensive literature implicating the enhancer-bound genes in tumor progression (see text). The two groups are functionally too divergent to be studied and presented within the same manuscript.

• *Figure 2G, We do see a decrease in H3K27ac at enhancers but there's also a decrease in binding at the promoter ITGB1, can the authors explain this result?*

The reviewer is correct. The slight decrease in the levels of H3K27ac following KMT2C/KD1 can either attributed to the fact that KMT2C binds both the enhancer and the promoter, or that the promoter is immunoprecipitated indirectly through enhancer/promoter interaction. The latter is quite possible in light of recent reports implicating KMT2C (and KMT2D) in long range interactions and promoter binding. This is now discussed in the context of Figure 3 (bottom of page 7 in the manuscript).

• *In the manuscript text, the authors mention controls, however the term is vague and can be specified (empty vector control, shRNA control, etc...)*

Throughout the manuscript, as controls we mean Scr. This has been clarified and explicitly stated throughout the manuscript.

• In Figure 2, how does the expression of KMT2C in HTB9 and T24 compared to other cancer cell lines? Is it considered highly expressed? Why were these cell lines chosen?

In general, cell lines were chosen on the basis of being WT for KMT2C and variable for p53 status. With respect to KMT2C expression, as expected cell lines expressed KMT2C at variable levels (see also comment below and Figure 4B). HTB9 and T24 specifically, are slightly above and below, respectively, of the mean (Figure 4B).

• In Figure 2, the authors showed knockdown of KMT2C in two independent lines (HTB9 and T24). However, in the rest of figure 2 RNA-seq/ChIP-seq analysis, was this done in comparing both cell lines? Or just HTB9? To demonstrate the broad transcriptional network of KMT2C, another independent cell line is required.

We apologize for the confusion. The RNA-seq and ChIP-seq experiments were performed only in HTB9 cells. Although our RNA-seq and ChIP-seq data in HTB9 cells indicate that a large number of genes could be regulated by KMT2C, we decided to focus our studies to those encoding proteins of the DNA damage response and DNA repair networks, which were in the top of the GO analysis list of processes affected by KMT2C knockdown. The positive correlation between KMT2C levels and the expression of DNA repair components was confirmed in many more cell lines, in clinical samples from the TCGA and in our own cohort of BC patients. Moreover, the KMT2C binding on these promoters is confirmed independently in breast cancer cells (Supplementary Figure S4).

• The results need p-values in Figure 3D.

This is correct. P values have been added.

• In Figure 4B, please label the type of cancers associated with these cell lines. Also among the other cell lines, H1792 cells did not decrease. Can the authors comment? Also are all these cell lines low in KMT2C expression?

All cell lines express variable but generally moderate levels of KMT2C, at least in comparison with other cell lines of the same cancer type. We have no explanation for the insufficient knockdown of KMT2C in H1792 and other cell lines. We decided to keep them more as controls because it strengthens our claims. In Figure 4B (now 4C), H1792 represent the last cell line in the row for each gene (ATM, ATR, BRCA1, BRCA2). It is quite evident that H1792 is the only cell line that does not show reduced expression of the four genes in the KD1 situation, supporting our claims that KMT2C controls their expression. We discuss this in the figure legend in the revised manuscript.

*• In Figure 5A (right), it is unclear which samples is compared to measure the significance. It looks like KD2 >5 foci is the same level as KD1, however KD2 has ** significance.*

All comparisons are pairwise against the Scr cells. Both the 1-5 foci and >5 foci have been compared to the respective Scr cells. The KD2 cells have more >5 foci than KD1 and many more than the Scr (the light grey bar is much taller in KD2 cells). Thus the two stars while KD1 has 1 star. What is not clear maybe is that the Y axis show the sum (stacked percentages) of 1-5 and >5 foci. We apologize for the confusion. We have included a more elaborate explanation in the legend.

• For Figure 6A, it is unclear what the representative image is? Is it KD1? If so, there's need to be a Scr control to compare it to.

Indeed it is a representative image from KD1 cells. What the graph is showing is the increased frequency of micronuclei. Other than that, micronuclei look the same irrespectively of the cell, only frequency changes. The pictures serve as representations of the abnormalities scored. Therefore, we considered showing an identical image from Scr control cells to be redundant and unnecessary use of Figure space. However, if the reviewer insists on presenting a representative image from Scr cells, we would be happy to do so.

• Figure 6E needs p-values of gain and loss.

This is a correct point. In the revised version, we have included p value < 0.0001. The actual values are 6.47e-12 and 2.57e-12 for KD1 and KD2 respectively.

- *Figure 6F panel is a bit messy and hard to interpret as a main figure and is essentially the same as 6E, please move to supplementary*

The reviewer is right about panel F being somewhat ineligible. We believe that this was due to its small size. We rearranged the panels and we hope that the reviewer is satisfied. We would rather keep this in the main figure because it is the only figure that shows which regions of what chromosomes are amplified or lost.

- *In Figure 7C, what was the concentration used for Olaparib and SCR-7? Please label concentration in legends or text.*

We apologize for omitting the drug concentrations. This has been fixed now.

2nd Editorial Decision

27 November 2018

Thank you for the submission of your revised manuscript to our editorial offices. We have now received the reports from the same three referees that have evaluated previously your study at The EMBO journal. As you will see, all referees now support the publication of your manuscript in EMBO reports. However, referee #3 has some further minor concerns or suggestions to improve the manuscript, we ask you to address in a final revised version of the manuscript.

Further, I have these editorial requests:

- I would suggest to change the title to:

The lysine-specific methyltransferase KMT2C/MLL3 regulates DNA repair components in cancer

- Please add up to 5 key words to the manuscript title page.

- Regarding the supplementary/additional data: The Expanded View format, which will be displayed in the main HTML of the paper in a collapsible format, has replaced the Supplementary information. You can submit up to 5 images as Expanded View. Please follow the nomenclature Figure EV1, Figure EV2 etc. The figure legends for these should be included in the main manuscript document file in a section called Expanded View Figure Legends after the main Figure Legends section. Thus, please select 5 figures from the 10 supplementary figures to be displayed as EV figures, and upload these as EV figures, and add their legends to the manuscript text.

- All the additional Supplementary material should be supplied as a single pdf file labeled Appendix. The Appendix file needs to include a table of content (TOC) on the first page, all figures and tables, and their legends. Please provide the file with page numbers, and also a TOC with page numbers. Please follow the nomenclature Appendix Figure Sx (Appendix Table Sx) throughout the text and also label the figures and tables according to this nomenclature. For more details please refer to our guide to authors: <http://embor.embopress.org/authorguide#manuscriptpreparation>

- Please remove all the text presently named 'List of Supplementary Materials' from the main manuscript text. This should become the TOC of the Appendix file. Only the legends of the 5 EV figures should remain in the main manuscript text file. If there are additional references for the Appendix, please move them there.

- Regarding data quantification and statistics, please check that where applicable the number "n" for how many independent experiments (biological replicates) were performed, the bars and error bars (e.g. SEM, SD) and the test used to calculate p-values is indicated in the respective figure legends. Please provide statistical testing where applicable. See: <http://embor.embopress.org/authorguide#statisticalanalysis>

- Please provide scale bars for all microscopic images. These are currently missing from Fig. 1C (UCC29), 5A, 5C, 5D, 6B/D, 8E, S6A/C, S7B and S8A/B.

- Please provide the accession number for the RNA-seq data (Short Read Archive) and add this to the manuscript text.

- As all the Western blots shown in the manuscript have been significantly cropped. We thus ask you to provide the of original source data for these, with the aim of making primary data more accessible and transparent to the reader. This source data will be published in a separate source data file online along with the accepted manuscript and will be linked to the relevant figure. Please submit the source data (scans of entire blots) of all the WBs shown (main figures, EV figures and Appendix figures) together with the revised manuscript. Please include size markers for scans of entire gels, label the scans with figure and panel number. Please send one PDF file per figure.

- Please provide the microscopic panels throughout in higher quality and better resolution. Please also provide the chromosome images throughout with less enhanced contrast, as unmodified as possible (many of these are currently overcontrasted).

- In figure S9 it seems some of the petri dishes have been rotated (e.g. the last ones in lanes KD1 and KD2). Could they all be shown in the same orientation? Is this a time course? Or are these not the same dishes in each lane? Please improve the legend to explain this better.

- Please format the references according to our journal style. See: <http://embor.embopress.org/authorguide#referencesformat>

- It seems that author Maria Tzetis is missing from the author contributions. Please add the missing information.

- Please find attached a word file of the manuscript text (provided by our publisher) with changes we ask you to include in your final manuscript text, and some queries (comments), we ask you to address. Please provide your final manuscript file with track changes, in order that we can see the modifications done.

- a letter detailing your responses to the final referee comments in Word format (.doc)
- a Microsoft Word file (.doc) of the revised manuscript text
- editable TIFF or EPS-formatted single figure files in high resolution (for main figures and EV figures)
- the Appendix file

In addition I would need from you:

- a short, two-sentence summary of the manuscript
- two to three bullet points highlighting the key findings of your study
- a schematic summary figure (in jpeg or tiff format with the exact width of 550 pixels and a height of not more than 400 pixels) that can be used as a visual synopsis on our website.

Referee #1:

The authors have done an impressive job addressing all the comments of the reviewers. I believe that this is an important contribution and deserved to be published soon.

Referee #2:

The authors have made a major effort to answer to the comments and I recommend publication.

Referee #3:

While it is well established that epigenetic dysregulation due to mutation of genes encoding

chromatin-modifying proteins occurs frequently in tumors, the mechanisms by which epigenetic dysregulation contributes to carcinogenesis are not well understood. Here the authors show that mutation of lysine-specific methyltransferase 2C results in reduced expression of key proteins involved in the DNA damage response and the repair of DNA double strand breaks by homologous recombination by increasing promoter methylation. As expected, the reduction in recombinational repair results in increased sensitivity to PARP inhibitors. Overall this study provides a novel link between epigenetic dysregulation due to defective lysine-specific methyltransferase 2C activity and reduced recombinational repair that is of general interest because of the clinical relevance. There are, however, a few minor concerns.

(i) The conclusion that cells with defective lysine-specific methyltransferase 2C activity are more dependent on PARP1/2 for DNA repair (stated in abstract and other places) is an overstatement. While these cells are likely to be more dependent upon PARP1-mediated alt NHEJ for the repair of DNA double strand breaks, the increased sensitivity to PARP inhibitors is also likely due to the increased number of replication-associated DNA double strand breaks caused by inhibition of PARP1-dependent repair of single strand breaks exceeding repair capacity.

(ii) The DSB repair field is not "largely unexplored" as stated by the authors on pg. 12. There are a large numbers of papers describing DSB repair pathway mechanisms, regulation of DSB repair pathway choice and altered DSB repair in cancer cells.

(iii) Is the graph in Fig. 7B showing chromosome fusions rather than ligation events?

2nd Revision - authors' response

17 December 2018

Referee #3:

While it is well established that epigenetic dysregulation due to mutation of genes encoding chromatin-modifying proteins occurs frequently in tumors, the mechanisms by which epigenetic dysregulation contributes to carcinogenesis are not well understood. Here the authors show that mutation of lysine-specific methyltransferase 2C results in reduced expression of key proteins involved in the DNA damage response and the repair of DNA double strand breaks by homologous recombination by increasing promoter methylation. As expected, the reduction in recombinational repair results in increased sensitivity to PARP inhibitors. Overall this study provides a novel link between epigenetic dysregulation due to defective lysine-specific methyltransferase 2C activity and reduced recombinational repair that is of general interest because of the clinical relevance. There are, however, a few minor concerns.

(i) The conclusion that cells with defective lysine-specific methyltransferase 2C activity are more dependent on PARP1/2 for DNA repair (stated in abstract and other places) is an overstatement. While these cells are likely to be more dependent upon PARP1-mediated alt NHEJ for the repair of DNA double strand breaks, the increased sensitivity to PARP inhibitors is also likely due to the increased number of replication-associated DNA double strand breaks caused by inhibition of PARP1-dependent repair of single strand breaks exceeding repair capacity.

The reviewer is absolutely correct on this issue: increased dependence KMT2C/KD cells to PARPi could also be due to unrepaired SSBs that generate DSBs. On the other hand, in KMT2C/KD untreated cells (no IR or cisplatin) the increased frequency of DSBs (H2AX foci) is due to HR deficiency (collapsed forks leading to DSBs; Figure 5A and EV2A). This explains the sensitivity of KD cells to olaparib even in the absence of exogenously-induced damage. This said, in the text we discuss the KMT2C/KD dependence upon PARP for "DNA repair" rather than "DSB repair". However, to acknowledge the role of PARP in SSB repair, we have included a sentence in the last paragraph of the Discussion section reading: Because PARP1/2 participate in the repair of single strand breaks (SSB; also induced during DNA replication), we cannot exclude the possibility that the increased sensitivity of KMT2C/KD cells to PARPi is also due to unrepaired SSBs which contribute to excess DNA damage.

(ii) The DSB repair field is not "largely unexplored" as stated by the authors on pg. 12. There are a

large numbers of papers describing DSB repair pathway mechanisms, regulation of DSB repair pathway choice and altered DSB repair in cancer cells.

The reviewer is correct. Maybe this was an overstatement. Thus, it has been removed from text.

(iii) Is the graph in Fig. 7B showing chromosome fusions rather than ligation events?

We count chromosome fusions and present each of them as one ligation event (in the text, they were reported as chromosome fusions and in figure legends and figures as ligation events). The reviewer, however, is right. For clarity purposes, we replaced the term ligation event with chromosome fusion in figures and figure legends.

Corresponding Author Name: Apostolos Klinakis

Journal Submitted to: EMBO reports

Manuscript Number: EMBOR-2018-46821